# FREEFUSE: MULTI-SUBJECT LORA FUSION VIA AUTO MASKING AT TEST TIME

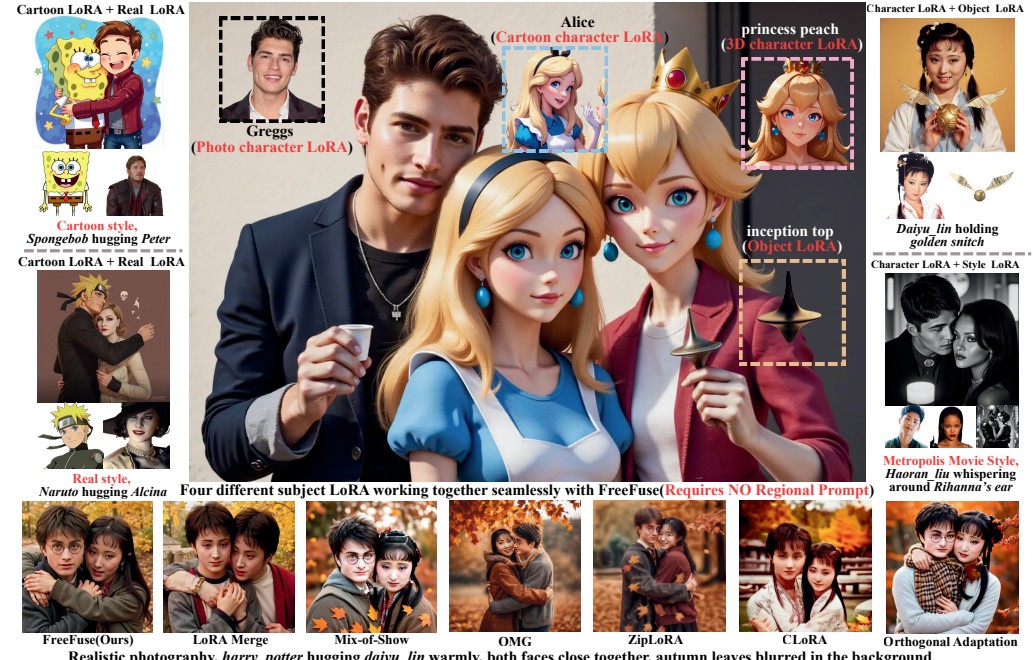

Figure 1: This paper proposes **FreeFuse**, a highly practical method that requires **no training**, **no modifications to existing LoRA models**, **no external models like segmentation models**, and **no user-defined prompt templates or region specifications**, yet fully unlocks the capability of large DiT models to generate high-quality multi-subject interaction images.

NEW

## ABSTRACT

This paper proposes FreeFuse, a novel training-free approach for multi-subject text-to-image generation through automatic fusion of multiple subject LoRAs. In contrast to existing methods that either focus on pre-inference LoRA weight merging or rely on segmentation models and complex techniques like noise blending to isolate LoRA outputs, our key insight is that context-aware dynamic subject masks can be automatically derived from cross-attention layer weights. Our analysis shows that constraining each LoRAs influence to its corresponding subject region via these masks effectively mitigates feature conflicts between LoRAs. FreeFuse demonstrates superior practicality and efficiency as it requires no additional training, no modification to LoRAs, no auxiliary models, and no user-defined prompt templates or region specifications. Alternatively, it only requires users to provide the LoRA activation words for seamless integration into standard workflows. Extensive experiments validate that FreeFuse outperforms existing approaches in both generation quality and usability under the multi-subject generation tasks. We will release the source code upon the official publication of the paper.

subjects

**Ziplora** performs well on the fusion tasks of style LoRA and subject LoRA, but is not suitable for multi-subject LoRA fusion.

**Mix-of-show** shows degraded generation quality in complex scenes, with attention to multi-hand issues and male hair length errors.

**OMG** heavily relies on the LoRA's redraw tendency aligning with the base model during the second-stage generation. Otherwise, subject features become inconsistent. For example, in stage one the male face is a full profile, while in stage two the LoRA-tuned model attempts to produce a three-quarter view, leading to feature confusion.

**Clora** performs well on joint inference of common subjects (such as cats and dogs), but is prone to collapse in complex multi-character scene.

**FreeFuse(Ours)** still performs well in multi-character and complex generation tasks (close-range interactions between characters).

Figure 2: An intuitive comparison of results, the prompt is ***harry-potter*** *tucking a flower in* ***daiyu-lin****s hair, both smiling warmly face-to-face.* Our method FreeFuse demonstrates significant advantages in generating complex character interaction scenes.

# 1 INTRODUCTION

Large-scale text-to-image (T2I) models such as FLUX.1-dev (Labs et al., 2025) (Labs, 2024) and HiDream (Cai et al., 2025) have demonstrated remarkable performance in general T2I tasks. To enhance their capability for personalized generation, Low-Rank Adaptation (LoRA) (Hu et al., 2022) has emerged as a preferred approach due to its precise fine-tuning quality and computational efficiency in both training and inference. LoRA also enables a simple way for multi-subject generation: As highly modular and portable modules, multiple subject LoRAs can be directly combined on the pretrained T2I models for generating multi-subject images. However, this straightforward approach can lead to significant performance degradation, with the appearance of feature conflicts and quality deterioration, making multi-subject LoRA fusion a challenging problem.

Prior works on multi-LoRA generation (Shah et al., 2024; Gu et al., 2023; Kong et al., 2024; Meral et al., 2024; Kwon et al., 2024) rely on designated techniques such as retraining, additional trainable parameters, external segmentation models or requiring users to provide template prompts or directly constrain the regions where LoRAs take effect, yet still struggle with multi-subject generation in complex scene (Fig. 2). To address the challenge of generating complex multi-subject scenes with multiple LoRAs, we analyzed the root cause of conflicts between subject LoRAs: during joint inference, they strongly compete in key regions such as faces. Based on this insight, we further conducted mathematical analysis and showed that constraining each subject LoRAs output to its target region via masks effectively mitigates feature conflicts. Our method, FreeFuse, consists of two stages. In the first stage, by addressing attention sink, exploiting the locality of self-attention, and applying patch-level voting, we obtain high-quality masks without retraining, LoRA modifications, auxiliary models, or prompt engineering. In the second stage, the extracted masks directly constrain LoRA outputs to the masked regions, avoiding the complex feature replacement (Gu et al., 2023) or noise blending strategies (Kong et al., 2024) used in prior work. In terms of efficiency, our method requires only a single step out of $n$ inference steps and a single attention block out of $m$ to get highly usable subject masks, offering a clear advantage over approaches that repeatedly update attention maps during inference (Meral et al., 2024). FreeFuse achieves high-quality, efficient multi-subject generation and can be seamlessly integrated into standard T2I workflows. In summary, our core contributions to the community include:

(1) An analysis of the cause of feature conflicts during joint inference with multi-subject LoRAs. We observe that the core issue is that, during joint inference, a subject LoRA not only influences its designated region but also tends to affect regions belonging to other subjects, leading to severe feature conflicts. Based on this finding, we mathematically analyze why mask-based LoRA output fusion can effectively alleviate such conflicts.

(2) A general solution for mitigating interference between conflicting LoRAs in DiT models, while preserving their original weights even in cases of overfitting. This solution isolates conflicts between LoRAs using masks automatically derived from attention maps and requires no trainable parameters, makes no modifications to LoRA modules, uses no auxiliary models, and does not rely on additional prompts from users for compatibility.

(3) A portable and highly efficient framework, FreeFuse, for multi-subject scene generation, Experimental results demonstrate that FreeFuse surpasses previous methods in both alleviating feature conflicts and enhancing image quality.

## 2 RELATED WORK

### 2.1 TEXT-TO-IMAGE DIFFUSION MODEL

In recent years, image generation models have advanced rapidly, evolving from early GAN-based models (Goodfellow et al., 2014) (Arjovsky et al., 2017) (Karras et al., 2019) (Karras et al., 2020) to U-Net-based diffusion models (Ronneberger et al., 2015) (Ho et al., 2020) (Song et al., 2020) (Rombach et al., 2022), and further to the widely adopted DiT-based diffusion models (Podell et al., 2023) (Peebles & Xie, 2023) (Esser et al., 2024) (Labs, 2024) (Cai et al., 2025) (Wu et al., 2025a). With the continuous growth of model size, training scale, and architectural improvements, large-scale DiT-based models such as FLUX.1-dev (Labs, 2024) have become leaders among open-source models, while also driving research into customized generation, local editing, and style transfer.

### 2.2 PERSONALIZED IMAGE GENERATION FOR DIFFUSION MODELS

Customized generation in diffusion models has been extensively studied. Textual inversion (Gal et al., 2022) methods encode rich semantic information into one or several text tokens through training. IP-Adapter (Ye et al., 2023),FLUX-Redux (Labs, 2024) and InstantID (Wang et al., 2024) instead train a generalizable module that directly takes one or more images and encodes their semantics into features aligned with the text or latent space. DreamBooth (Ruiz et al., 2023) introduces new concepts by fine-tuning diffusion network weights. With the wide adoption of LoRA (Hu et al., 2022) as an efficient fine-tuning method, fine-tuning open-source diffusion models with LoRA for customized generation has become a common choice among community users. Numerous works further improve LoRA or its training strategies, such as LyCORIS (Yeh et al., 2023), QLoRA (Dettmers et al., 2023), ED-LoRA (Gu et al., 2023), and SD-LoRA (Wu et al., 2025b), but LoRA itself remains the most widely used solution.

### 2.3 MULTI-LORA BASED MULTI-CONCEPT GENERATION

This work focuses on multi-concept generation through joint inference with multiple LoRAs. The performance degradation caused by multi-LoRA inference was first widely studied in large language models (LLMs), where researchers observed significant quality drops when integrating multiple LoRAs. Various approaches were proposed, including clustering LoRAs in advance (Zhao et al., 2024), introducing gating networks (Wu et al., 2024), and retraining with conflict-mitigation objectives (Feng et al., 2025). In text-to-image models, similar directions have been explored. Methods such as ZipLoRA (Shah et al., 2024) and K-LoRA (Ouyang et al., 2025) fuse multiple LoRAs before inference, achieving notable success in style transfer but limited performance in multi-concept generation. Multi-LoRA (Zhong et al., 2024) further proposed switch and composite strategies for conflict mitigation during inference, showing promising results in character-object compositions but struggling in multi-character scenarios.

Table 1: Method feature comparison between recent works.

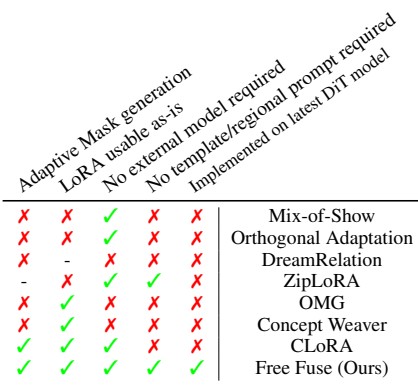

| Adaptive Mask generation | LoRA usable as-is | No external model required | No template/regional prompt required | Implemented on latest DiT model | |
|---|---|---|---|---|---|
| ✗ | ✗ | ✓ | ✗ | ✗ | Mix-of-Show |
| ✗ | ✗ | ✓ | ✗ | ✗ | Orthogonal Adaptation |
| ✗ | - | ✗ | ✗ | ✗ | DreamRelation |
| - | ✗ | ✓ | ✓ | ✗ | ZipLoRA |
| ✗ | ✓ | ✗ | ✗ | ✗ | OMG |
| ✗ | ✓ | ✗ | ✗ | ✗ | Concept Weaver |
| ✓ | ✓ | ✓ | ✗ | ✗ | CLoRA |
| ✓ | ✓ | ✓ | ✓ | ✓ | Free Fuse (Ours) |

For multi-character tasks, OMG (Kong et al., 2024) introduces an auxiliary model to localize character regions and applies noise blending, but heavily relies on the LoRAs redraw tendency aligning with the base model during the second-stage generation. Mix-of-Show (Gu et al., 2023) and Orthogonal Adaptation (Po et al., 2024) requires retraining the LoRA and manually specifying its spatial constraints. Some methods, such as DreamRelation (Shi et al., 2025), achieve precise character and relation generation through extensive fine-grained control, but this increases the difficulty for users to employ the method. Concept Weaver (Kwon et al., 2024) mitigates

NEW

NEW

(a) When two subject LoRAs are jointly inferred, they often exhibit severe competition in the critical regions of each subject. We compare the cosine similarity of their latent-space outputs in denoising step 6, 13, 19 and 26, with visualizations confirming this effect. Notably, **Alcina Dimitrescu** and **Spock**'s LoRAs strongly interfere in each other's facial regions during the inference, leading to Alcina's pale facial traits invading Spock, while her own face acquires flesh tones due to Spocks feature intrusion.

(b) The attention maps of the Spock region show clear locality.

Figure 3: Conflicts Analysis

this issue with Fusion Sampling, but still heavily relies on segmentation quality. CLoRA (Meral et al., 2024) leverages attention maps to derive concept masks, yet requires template prompts as a basis for mask extraction, and its performance drops in complex multi-concept scenes. See Fig. 2. Moreover, except for K-LoRA, the above methods were implemented only on earlier U-Net-based models, while the multi-lora based multi-concept generation capability of more advanced DiT models remains largely unexplored. We compared our method with other methods based on their characteristics, as shown in Table 1, demonstrating that our method exhibits significantly superior usability compared to other approaches.

## 3 ANALYSIS

In this section, we demonstrate a major cause of feature conflicts in multi-subject joint inference: the intense competition among LoRAs in key subject regions, such as faces. We then utilize an intuitive solution, restricting each LoRA to operate only within the region of its corresponding concept, and show how this serves as a good approximation that effectively mitigates feature conflicts among subject LoRAs.

### 3.1 INTERFERENCE BETWEEN SUBJECT LoRAs DURING JOINT INFERENCE

One would naturally expect that, when multiple subject LoRAs are jointly applied to a diffusion model, each should influence only its corresponding subject. However, in practice this is not the case. Examination of the latent space reveals strong competition among LoRAs, particularly in the most distinctive regions of each subject, such as character faces. As illustrated in Fig. 3a, this competition results in severe feature conflicts and confusion.

### 3.2 MASKING LoRA OUTPUTS FOR EFFECTIVE SUBJECT FEATURE PRESERVATION

Consider $N$ distinct subjects $\{S_1, \ldots, S_N\}$ generated via specific LoRA adapters $\{\Delta\theta_1, \ldots, \Delta\theta_N\}$, corresponding to spatial regions $\{R_1, \ldots, R_N\}$ in the latent space. Directly merging all the LoRAs involves a naive summation of all LoRA outputs. This can introduce severe feature interference, as a token at position $p \in R_k$ is simultaneously perturbed by conflicting outputs from unrelated subjects.   FIX

To enforce subject disentanglement, we propose a spatial masking strategy. For any token position $p$, we strictly retain the contribution of the corresponding LoRA while suppressing others:   FIX

$$h'_p = h_p + \sum_{i=1}^{N} \mathbb{I}(p \in R_i) \cdot \Delta\theta_i(x_p), \tag{1}$$

where $\mathbb{I}(\cdot)$ is the indicator function. This ensures that the feature update at region $R_k$ is exclusively governed by $\Delta\theta_k$.   FIX

A potential concern is that LoRA features from disjoint regions $R_{j \neq k}$ might still propagate into $R_k$ through the global aggregation of the self-attention mechanism:   FIX

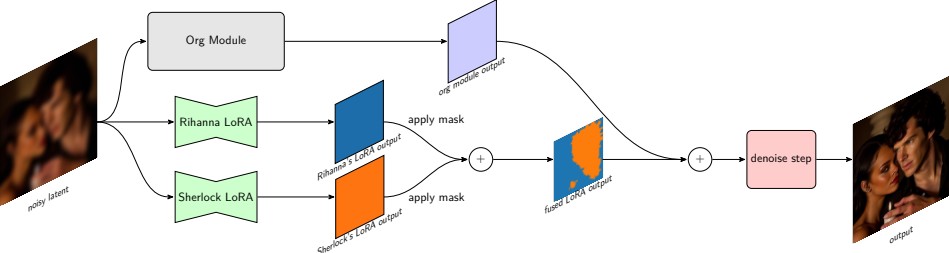

(a) **First stage.** We extract cross-attention maps to localize the regions associated with each subject prompt. The attention sink issue is mitigated, and top-k elements are used to derive self-attention maps, whose stronger locality further enhances usability. The predicted image is then segmented into superpixels, with block-level voting to assign ownership, producing reliable subject masks.

(b) **Second stage.** The masks are repeatedly applied during inference, constraining each LoRA to its designated region and mitigating feature conflicts among them.

Figure 4: **Pipeline.** Our pipeline consists of two stages: the first derives subject masks from attention maps, and the second applies these masks to LoRA outputs, ensuring that each LoRA only operates within its corresponding subject region.

$$\text{Attn}(Q, K, V)_p = \sum_q A_{p,q} V_q, \tag{2}$$

where $V_q$ carries features from remote regions. However, we leverage the intrinsic ***spatial locality*** widely observed (Raghu et al., 2022; Caron et al., 2021; Helbling et al., 2025) in DiT models, as shown in Fig. 3b. The attention map $A$ exhibits a strong diagonal dominance, where tokens in region $R_k$ predominantly attend to other tokens within the same region:

$$\sum_{q \in R_k} A_{p,q} \gg \sum_{q \notin R_k} A_{p,q}, \quad \forall p \in R_k. \tag{3}$$

Consequently, cross-region information leakage is negligible. By applying spatial masks, we effectively isolate the feature spaces of different subjects, ensuring that $S_k$ is synthesized solely based on $\Delta\theta_k$ without contamination from the aggregation process.                          FIX

## 4 METHOD

As illustrated in Fig. 4, FreeFuse adopts a two-stage pipeline. In the first stage, the subject mask is automatically calculated through cross attention map from only one layer and one step. In the second stage, the masks are repeatedly applied during inference. Below we introduce the key steps for subject mask calculation.

## 4.1 CROSS ATTENTION MAP COMPUTATION AND ATTENTION SINK HANDLING

We compute cross attention maps between text queries and image keys through standard scaled dot-product attention:

$$\mathbf{A}_{\text{cross}} = \text{softmax}\left(\frac{\mathbf{Q}_{\text{text}}\mathbf{K}_{\text{img}}^T}{\sqrt{D}}\right), \tag{4}$$

where $\mathbf{Q}_{\text{text}} \in \mathbb{R}^{B \times N_{\text{text}} \times D}$ and $\mathbf{K}_{\text{img}} \in \mathbb{R}^{B \times N_{\text{img}} \times D}$ are the text queries and image keys respectively, $B$ means batch size, $N$ means sequence length.

However, raw attention maps often exhibit the "attention sink" phenomenon where boundary pixels accumulate excessive attention weights. To address this, we apply a heuristic filtering mechanism that combines Top-K thresholding with spatial edge detection:

$$\mathcal{M}_{\text{topk}}(i,j) = \mathbb{I}[\mathbf{A}(i,j) \geq \tau_k], \quad \mathcal{M}_{\text{edge}}(i,j) = \mathbb{I}[(i,j) \in \mathcal{E}], \quad \mathcal{M}_{\text{handle\_sink}} = \neg(\mathcal{M}_{\text{topk}} \wedge \mathcal{M}_{\text{edge}}), \tag{5}$$

where $\tau_k$ is the $k$-th largest attention value with $k = \lfloor N_{\text{img}} \times p \rfloor$, in practice, we take $p$ as 1%, $\mathcal{E}$ represents edge pixel regions, and $\mathbb{I}[\cdot]$ is the indicator function. The filtered attention map is then normalized:

$$\tilde{\mathbf{A}} = \frac{\mathbf{A} \odot \mathcal{M}_{\text{handle\_sink}}}{\sum_j (\mathbf{A} \odot \mathcal{M}_{\text{handle\_sink}})_{ij}}. \tag{6}$$

## 4.2 LoRA ACTIVATION WORD ATTENTION MAP DERIVATION

Given LoRA activation words $\{w_1, w_2, \ldots, w_L\}$ with token position sets $\{\mathcal{I}_1, \mathcal{I}_2, \ldots, \mathcal{I}_L\}$, we first extract the cross-attention map for each LoRA by averaging over its corresponding token positions:

$$\mathbf{M}_l = \frac{1}{|\mathcal{I}_l|} \sum_{idx \in \mathcal{I}_l} \tilde{\mathbf{A}}[idx, :]. \tag{7}$$

Cross-attention maps from different LoRAs often exhibit mutual interference, while self-attention maps demonstrate stronger locality, leading to more cohesive attention patterns. We identify the most salient regions by selecting the top 1% pixels from the cross-attention map:

$$\mathcal{T}_{1\%} = \text{TopK}(\mathbf{M}_l, K = \lfloor N_{\text{img}} \times 0.01 \rfloor). \tag{8}$$

The final attention map leverages self-attention from these salient regions:

$$\mathbf{M}_l^{\text{self\_attn}} = \frac{1}{|\mathcal{T}_{1\%}|} \sum_{i \in \mathcal{T}_{1\%}} \mathbf{A}_{\text{self}}[i, :], \tag{9}$$

where $\mathbf{A}_{\text{self}} \in \mathbb{R}^{N_{\text{img}} \times N_{\text{img}}}$ is the self-attention map computed between image tokens, and $\mathbf{M}_l^{\text{self\_attn}}$ represents the enhanced spatial attention distribution of the $l$-th LoRA activation word.

## 4.3 SUPERPIXEL-BASED ENSEMBLE MASKING

To address the hole artifacts that arise from pixel-wise competition between LoRA attention maps, we introduce a superpixel-based ensemble approach. At designated denoising steps, we utilize the predicted sample $\mathbf{x}_0$ to generate spatially coherent regions via SLIC superpixel segmentation:

$$\mathcal{R} = \text{SLIC}(\mathbf{x}_0, n_{\text{segments}}, \text{compactness}, \sigma). \tag{10}$$

In practice, $n_{\text{segments}}$ is taken as the square root of the target image area and compactness is taken as 10. For each superpixel region $r_j \in \mathcal{R}$, we compute the aggregated attention score for each LoRA:

$$s_{l,j} = \sum_{(u,v) \in r_j} \mathbf{M}_l^{\text{self\_attn}\uparrow}(u,v), \tag{11}$$

Table 2: Average, 10-Pass score for DINOv3, DreamSim(1 − Score), LVFace, HPSv3 and Vision Language Model. For all metrics, the higher, the better.

| | | LoRA Merge | Orthogonal Adaptation | ZipLoRA | OMG | Mix-of-Show | CLoRA | **Ours** |
|---|---|---|---|---|---|---|---|---|
| DINOv3 | Avg. | 0.5314 | 0.5294 | 0.4781 | 0.4457 | 0.5284 | 0.4452 | **0.5397** |
| | 10-Pass. | 0.5946 | 0.5361 | 0.5256 | 0.5045 | 0.5789 | 0.4953 | **0.5949** |
| DreamSim | Avg. | 0.7242 | 0.7181 | 0.6648 | 0.6292 | 0.7324 | 0.6413 | **0.7368** |
| | 10-Pass. | 0.7683 | 0.7797 | 0.7187 | 0.7025 | 0.7921 | 0.7037 | **0.8052** |
| LVFace | Avg. | 0.2876 | 0.3376 | 0.2037 | 0.2179 | **0.3430** | 0.1837 | 0.3302 |
| | 10-Pass. | 0.3698 | 0.4493 | 0.2720 | 0.3018 | 0.4417 | 0.2625 | **0.4685** |
| HPSv3 | Avg. | 9.128 | 6.771 | 9.024 | 9.052 | 6.868 | 5.526 | **10.63** |
| | 10-Pass. | 10.71 | 8.693 | 10.92 | 10.80 | 8.644 | 9.383 | **12.25** |
| VLM Score | | 51.94 | 58.22 | 49.97 | 53.02 | 57.74 | 23.56 | **74.03** |

where $\mathbf{M}_l^{\text{self\_attn}\uparrow}$ denotes the upsampled attention map to match the image resolution. The winning LoRA for region $r_j$ is determined by $l^* = \arg\max_l s_{l,j}$, and the final binary mask for the $l$-th LoRA is constructed as:

$$\mathbf{F}_l(u, v) = \begin{cases} 1, & \text{if } (u, v) \in r_j \text{ and } l^* = l, \\ 0, & \text{otherwise.} \end{cases} \quad (12)$$

This superpixel-based voting mechanism ensures spatially coherent masks while preserving fine-grained regional boundaries. Our empirical study shows that it is unnecessary to compute attention maps at every layer or denoising step. For instance, in the common 28-step inference of the FLUX.1-dev model, extracting subject masks solely from the attention of the 17th Double Stream Block at the 6th denoising step is sufficient, yielding a substantial gain in efficiency.

## 5 EXPERIMENTS

From the experiments, our method is evaluated against prior approaches in the following aspects:

(1) Ability to best preserve subject characteristics in complex scenes.

(2) Ability to generate images with quality closest to the pretraining data.

(3) Robustness in adhering to complex prompts.

(4) Alignment with human preference in terms of lighting, details, realism, and artifact-free generation.

we use direct LoRA joint inference as our baseline and compare against ZipLoRA (Shah et al., 2024), OMG (Kong et al., 2024), Mix-of-Show (Gu et al., 2023), Orthogonal Adaptation (Po et al., 2024), and CLoRA (Meral et al., 2024) as comparative methods. Since the five methods involve four     NEW
different pretrained text-to-image models, it is difficult to obtain subject LoRAs from the community that are compatible with all of them. To ensure fairness in comparison, We prepared identical 5-character LoRAs for each method pipeline, resulting in 20 LoRAs in total, as conflicts between character LoRAs are often the most severe and can effectively reflect each method's actual capability in mitigating inter-LoRA feature conflicts. Each LoRA was trained following the optimal training method recommended by the respective method's base pipeline and used exactly the same datasets. We prepared 50 prompt sets as shown in Appendix A, all involving character interactions with many incorporating complex actions and environments to thoroughly examine each method's performance on complex generation tasks.

### 5.1 QUANTITATIVE RESULTS

We designed four evaluation metrics to assess method performance. First, following OMG, we employ a face recognition model to evaluate how well each method preserves character-specific features. But unlike their use of arcface (Deng et al., 2019), we employed the current state-of-the-art LVFace (You et al., 2025) for facial similarity scoring. This metric effectively addresses evaluation objective (1). We also use DINOv3 (Siméoni et al., 2025) to detect subject regions in the generated images and measure their feature similarity with training images, which effectively reflects evaluation objective (2). We observed that DINOv3 often yields high similarity for artifact-heavy images. Hence, we additionally use DreamSim (Fu et al., 2023), which better aligns with human

(a) Qualitative Comparison: Each row uses the same prompt. For all methods except LoRA Merge, we report the result with the highest IDA among 10 samples. For LoRA Merge, we use the same seed as FreeFuse to highlight our improvements over direct inference.

(b) More Qualitative results: Our method excels in image details, lighting, character quality, and realism, and effectively generates complex interactions such as physical contact that prior methods struggle with.

Figure 5: Qualitative results

preferences, to evaluate objective (2). We further employ HPSv3 (Ma et al., 2025), a state-of-the-art human preference alignment model proven highly effective in reinforcement learning (Xue et al., 2025), to evaluate the image quality and instruction-following ability of each methods outputs. This metric effectively addresses evaluation objectives (3) and (4). Additionally, given the rapid advancement in Vision Language Models, we defined VLM scoring that evaluates across three dimensions: character consistency (50 points), prompt consistency (25 points), and image quality (25 points). The full prompt is shown in Appendix B. We use Gemini-2.5 (Comanici et al., 2025) as the scoring model. During testing, we paired the 5 character LoRAs pairwise to form 10 pairs, generating results from 10 seeds [42,52] for each prompt, resulting in each method generating 5000 images. For each method, we calculated both global averages and 10-Pass averages (taking the best result from 10 outputs per prompt for averaging). Our final results are shown in Table 2. For the LVFace-AVG metric, our score is slightly lower than Mix-of-Show, which relies on user-specified Rectangular regions to restrict LoRAs' outputs and thus avoids detection errors. In contrast, our adaptive masks may occasionally misalign but better capture complex subject interactions, leading to superior performance in 10-Pass tasks. Across other metrics, our approach outperforms the baselines and competing methods, demonstrating clear advantages in image quality, character feature preservation, and alignment with human preferences.

To demonstrate the scalability of our approach, we further evaluate it on a broad set of community-trained LoRAs. We collect 35 character LoRAs and 6 object LoRAs from Civitai, each with over 300 downloads to ensure practical relevance. We test configurations ranging from 1 character + 1 object to 4 characters + 1 object and report DINOv3, DreamSim, and LV-

Table 3: Experiment for exploring scalability.

| | methods | 1char+1obj | 2char | 3char+1obj | 4char+1obj |
|---|---|---|---|---|---|
| DINOv3 | Ours | **0.5869** | **0.5785** | **0.5296** | **0.5041** |
| | LoRA Merge | 0.5747 | 0.5295 | 0.5130 | 0.5000 |
| DreamSim | Ours | **0.7966** | **0.7620** | **0.7123** | **0.6897** |
| | LoRA Merge | 0.7453 | 0.7380 | 0.6543 | 0.6249 |
| LVFace | Ours | **0.5046** | **0.4076** | **0.3219** | **0.2772** |
| | LoRA Merge | 0.4824 | 0.3681 | 0.2270 | 0.1913 |

Face metrics in Table 3, with corresponding visual results provided in Fig. 6, more results are shown in Fig 11 and Fig 12. The results show that our method scales well: although performance gradually declines as the number of LoRAs increases, it is important to note that our approach requires no regional prompts to achieve the visual results shown. In contrast, prior methods capable of partially handling compositions with more than three characters, such as Mix-of-Show, DreamRelation, and Orthogonal Adaptation, depend on regional prompts to explicitly constrain where each LoRA can act. Our method, however, relies entirely on the models own scene understanding, requiring no additional user guidance. Once the number of LoRAs integrated into the model exceeded five, we observed that most failures were caused by inaccurate extracted masks. We attribute this to the inherent difficulty the base model already faces in understanding multi-person interaction scenes; as its scene comprehension degrades, the quality of the extracted masks also deteriorates. Detailed analysis can be found in Appendix G.

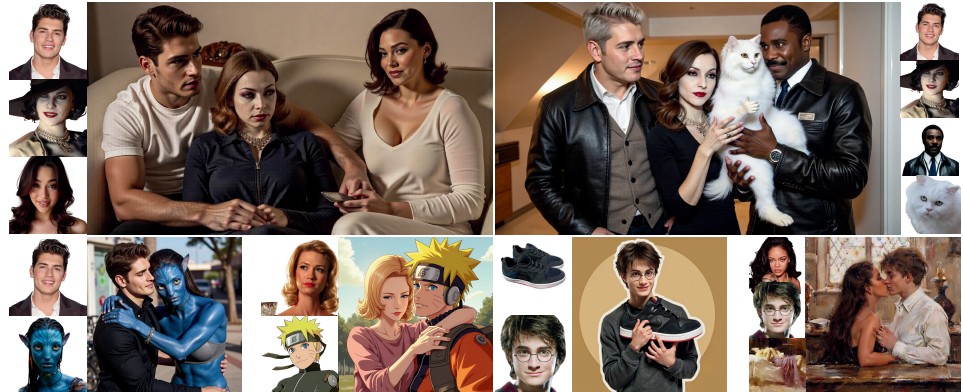

Figure 6: The visualizations show that our method can produce strong results even in scenes involving multiple characters with complex interactions. It also enables seamless fusion between characters and objects of various styles (e.g., cartoon and photorealistic), while remaining highly compatible with style LoRAs, simply applying an all-ones mask for style LoRAs is sufficient.

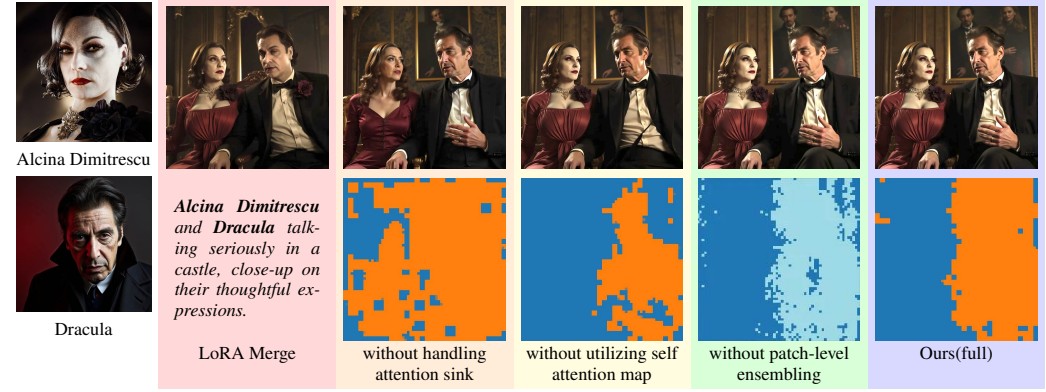

Figure 7: Ablation studies demonstrate that each step of our method is essential for producing highly usable subject masks.

## 5.2 QUALITATIVE RESULTS

The qualitative generation results are illustrated in Fig. 5. We present additional qualitative results in Appendix H. Our method shows advantages in image quality, instruction following, and subject feature preservation.

## 5.3 ABLATION STUDY

The success of our approach relies on a key factor: the accurate generation of high-quality object masks. We analyze the effects of removing different components, namely attention sink handling, the use of self-attention maps, and block-level voting. As shown in Fig. 7, omitting attention sink handling often causes one LoRA to over-focus on sink elements, allowing another LoRA to dominate most regions. Without self-attention maps, the extracted masks exhibit severe cross-intrusion. Without block-level voting, the masks contain numerous holes. All of these issues ultimately degrade the final generation quality.

To further demonstrate the superiority of our attention map extraction method, we analyze the difference in masks extracted at various timesteps. The results shown in Fig.8b indicate that extracting the mask only once during the initial steps of the denoising process offers both computational efficiency and superior accuracy. This is because the diffusion model typically establishes the primary image composition in the early denoising steps, with subsequent steps dedicated to fine-tuning, leading to masks derived from later steps often lacking structural coherence. We also conducted an ablation study on several hyperparameters, confirming that our current parameter settings are optimal. For the

future open-source release, we will provide a flexible interface allowing users to rapidly experiment with different hyperparameter configurations. Furthermore, our method exhibits an advantage over other mask extraction techniques, such as the one described in Self-Guidance (Epstein et al., 2023) and Readout-Guidance (Luo et al., 2024). The latter's extracted masks suffer from either excessive overlap or insufficient area (when employing a hard strategy), or they cluster narrowly around 0.5, effectively degrading the mask into a simple LoRA merge with a weight of 0.5, as shown in Fig.8a.

NEW

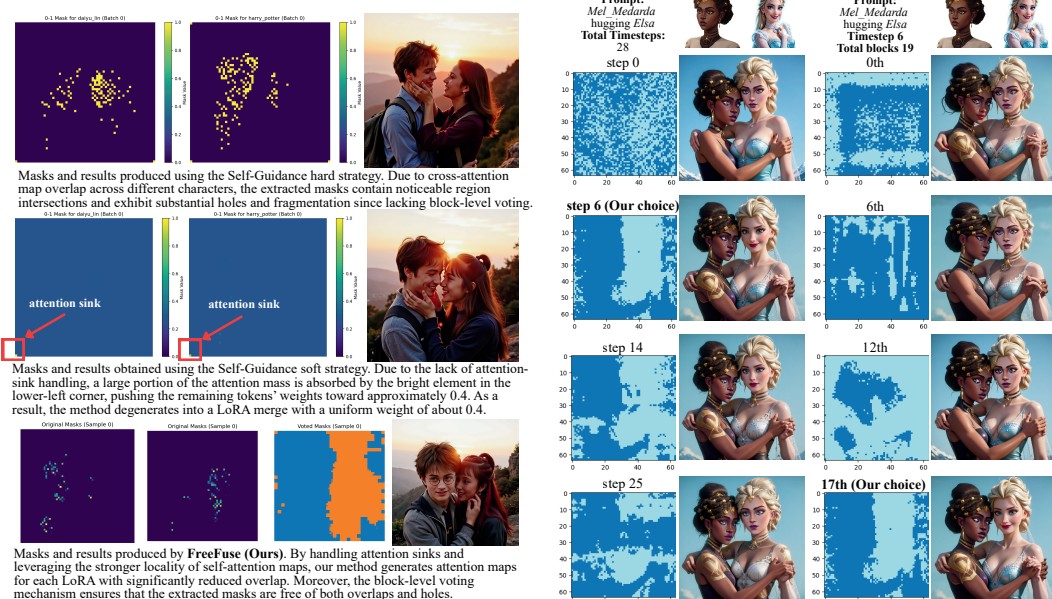

(a) Compared with methods that extract masks using fixed thresholds, or those that neither address attention sinks nor leverage self-attention maps, our approach demonstrates clear superiority.

(b) Ablation study further shows that extracting attention maps at the 6th/28 denoising step and the 17th double-stream block not only offers computational efficiency but also achieves the highest accuracy.

Figure 8: The ablation results indicate that our masking extraction strategy and hyperparameter choices provide clear advantages in both accuracy and efficiency compared with alternative methods.

NEW

## 6 CONCLUSION

We present FreeFuse, a highly practical multi-concept generation method designed to mitigate conflicts in multi-LoRA joint inference. Our analysis shows that restricting each LoRAs influence to its corresponding subject region via masks effectively mitigates feature conflicts between LoRAs. We leverage attention sink handling and self-attention maps with superpixel-based block voting, deriving high-quality subject masks from low-quality cross-attention maps. Our approach introduces no trainable parameters, requires no auxiliary models beyond the baseline, and avoids burdensome region masks or template prompts. Experiments demonstrate that FreeFuse achieves superior subject fidelity, prompt adherence, and generation quality in complex scenarios and tasks.

**Limitations and future work**. As shown in Appendix G, generating scenes with four or more characters remains challenging. Our method can produce artifacts due to limitations in the models inherent scene understanding, and potential solutions are discussed in Appendix G.

FIX

## REPRODUCIBILITY STATEMENT

Our method is built on the open-source FLUX-1.dev model, with implementation based on Huggingface Diffusers. The code and LoRA used will be released upon the official publication of this paper.

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

## A  EVALUATION PROMPTS

The prompts used in our evaluation are more challenging than those in prior work, including requirements for close subject interactions (e.g., hugging, kissing, caressing a face, whispering, tending a wound), complex actions (e.g., pillow fights, arm wrestling, eating pizza), and intricate lighting conditions (e.g., faces illuminated by a campfire or lantern).

---

**Evaluation Prompts**

1. <A> teaching  guitar, both sitting close together, their faces near as <A> guides 's fingers on the strings.

2. <A> kissing  tenderly in a quiet classroom, their faces close under soft afternoon light.

3. <A> holding 's face gently, both smiling after climbing a mountain, sunset light on their cheeks.

4. <A> whispering into 's ear, their faces almost touching, candlelight revealing 's expression.

5. <A> and  laughing together, faces dusted with flour as they bake a cake side by side.

6. <A> hugging  warmly, both faces close together, autumn leaves blurred in the background.

7. <A> and  sitting shoulder to shoulder by the fireplace, faces lit by its warm glow.

8. <A> carefully wrapping 's injured hand, both watching each others expressions closely.

9. <A> and  sharing headphones, leaning their heads together, faces relaxed as they listen to music.

10. <A> carrying  playfully, both laughing, their faces captured in a close, joyful moment.

11. <A> catching , both looking at each others faces, smiling in relief on the ice.

12. <A> and  painting, cheeks smeared with color, smiling at each other over the canvas.

13. <A> showing  a photo, both faces close as they look at the album together.

14. <A> gently cupping s face, their foreheads almost touching, eyes filled with tenderness.

15. <A> and  looking up together at the viewer, smiling softly, fairy lights reflecting in their eyes.

16. <A> handing cocoa to , both smiling warmly at each other, close by the fire.

17. <A> and  grinning face-to-face in the middle of a playful arm-wrestling match.

18. <A> pointing at the stars,  watching <A>s face with amazement.

19. <A> and  paddling, both faces determined, close-up of their focused expressions.

20. <A> guiding 's hands with care, their faces close together as they roll sushi.

21. <A> and  staring each other down across the table, intense eye contact filling the room.

22. <A> and  laughing face-to-face while kneeling by a sandcastle.

23. <A> adjusting 's bowtie, both faces inches apart, smiling shyly.

24. <A> holding up an artifact for , their faces close as they study it curiously.

25. <A> and  laughing mid-pillow fight, close-up of their faces among flying feathers.

26. <A> and  practicing dance steps, tangled and laughing, faces flushed with joy.

27. <A> performing a trick, 's amazed face in the foreground.

28. <A> and  eating pizza, close-up of them laughing together on the rooftop.

29. <A> and  planting flowers, smiling at each other, dirt smudges on their cheeks.

30. <A> helping  with armor, both concentrating on each others faces.

31. <A> handing  an apple, both laughing, their faces close together.

32. <A> and  talking seriously on the swings, close-up on their thoughtful expressions.

33. <A> and  leaning over a map, faces illuminated by the lantern glow.

34. <A> pushing  on the swing, both laughing, close-up on their happy faces.

35. <A> and  mid-tango, faces close with passionate expressions.

36. <A> showing  the glowing sword, their faces lit by the forges light.

37. <A> and  side by side on the couch, screen glow on their focused faces.

38. <A> and  assembling furniture, faces frustrated but laughing together.

39. <A> and  steadying the ladder, both faces anxious yet determined.

40. <A> and  sharing a secret glance, their eyes meeting in the crowded room.

41. <A> measuring  for a suit, both faces close and serious.

42. <A> and  roasting marshmallows, laughing as the firelight glows on their faces.

43. <A> showing  a bubbling potion, both gazing at each other in fascination.

44. <A> and  clinking glasses, their smiling faces framed by the Paris skyline.

45. <A> reading a story,  resting their head close, listening intently.

46. <A> bumping into , both kneeling to gather papers, surprised faces close together.

47. <A> and  sparring, close-up of their intense expressions and focused eyes.

48. <A> tucking a flower in s hair, both smiling warmly face-to-face.

49. <A> and  chasing fireflies, faces glowing in the jars soft light.

50. <A> and  back-to-back, turning to glance at each other with trust.

---

## B   PROMPT FOR VLM SCORING

```
You are an image quality evaluator specializing in character generation
    and image quality assessment.
Please evaluate the quality of the last image (the generated image)
    based on the following criteria:

Reference images: The first {len(reference_images)} images show
    reference characters <A> and  that should appear in the
    generated image.
Target image: The last image is the generated image that should be
    evaluated.
Generation prompt: "{prompt_text}"

Evaluation criteria (total 100 points):
1. Character presence and clarity (50 points): Both characters from the
     reference images appear in the target image with clear and
    recognizable features.
2. Prompt adherence (25 points): The generated image follows the
    requirements described in the prompt.
3. Image clarity and quality (25 points): The image is clear, not
    blurry, and free of artifacts.

Please provide:
1. Detailed analysis for each criterion
2. Score for each criterion (out of the maximum points)
3. Total score (sum of all criteria scores)
4. Brief reasoning for the scores

Format your response as:
Character Analysis: [your analysis]
Character Score: [0-50]
Prompt Analysis: [your analysis]
Prompt Score: [0-25]
Clarity Analysis: [your analysis]
Clarity Score: [0-25]
Total Score: [0-100]
Reasoning: [brief explanation]
```

## C   THE USE OF LARGE LANGUAGE MODELS

In writing, we used a large model to check and correct grammatical errors. In experiments, the large model was employed to evaluate the final generated images, and its scores served as a metric in our quantitative analysis.

## D  IMPLEMENTATION DETAILS

Our method is implemented on the FLUX.1-dev model, with the code built on Huggingface Diffusers (hunggingface, 2025). In the standard 28-step inference process, we do not intervene during the first 6 steps. At step 6, the subject mask is extracted by computing the Attention Map from the 17th double_stream_block. For superpixel-level voting, n_segments is set to the square root of the total image pixels. During the remaining denoising steps, each LoRA output is multiplied by this mask until inference completes. Experiments were conducted on a single NVIDIA L20 GPU with 48GB VRAM, achieving an average inference time of 34s, please refer to Appendix F for runtime comparison.                                                            FIX

The LoRAs used in our experiments were trained with the Aitoolkit (ostris, 2025) framework. For each character, 15 high-quality images covering multiple angles and diverse outfits were collected as the training dataset. Gemini-2.5 was used to generate prompts, and each LoRA was trained on the corresponding baseline until convergence.

## E  ADAPTATION TO OTHER DiT MODELS: TAKING SD3.5 AS EXAMPLE

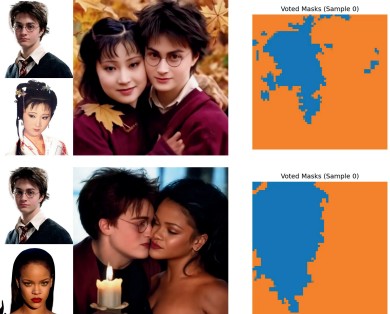

Table 4: Experiment for transfering FreeFuse to Stable Diffusion 3.5.

|  | DINOv3 | Dreamsim | LVFace | HPSv3 | VLM Score |
|---|---|---|---|---|---|
| sd3.5 | 0.4530 | 0.6841 | 0.2528 | 6.279 | 52.38 |
| sd3.5 + FreeFuse | **0.4883** | **0.7169** | **0.3486** | **7.24** | **61.71** |

Figure 9: Qualitative Results for FreeFuse on Stable Diffision 3.5

To demonstrate that our method transfers well to other DiT-based architectures, we implemented it on Stable Diffusion 3.5. As shown in Fig. 9 and Table 4, our approach exhibits strong generalization on this model as well. The entire migration required only replacing the 17th double-stream block used in Flux with the 25th transformer block of Stable Diffusion 3.5.                          NEW

## F  RUNTIME ANALYSIS

Table 5: Runtime Comparison.

| Method | Time | Memory |
|---|---|---|
| Flux LoRA Merge | 28s | 38G |
| Ours | 34s | 38G |
| SDXL LoRA Merge | 8s | 7G |
| CLoRA | 42s | 25G |
| OMG | 76s | 30G |
| Ours (Compute Every Timestep) | 44s | 38G |

Table 6: Performance Comparison: Baseline vs Ours.

| Configuration | LoRA Merge | Ours | Extra Time/Mem Required |
|---|---|---|---|
| bf16 | 28s / 38G | 34s / 38G | 21% / 0% |
| fp8 | 36s / 27G | 41s / 27G | 14% / 0% |
| fp8 + cpu_offloading | 48s / 14G | 59s / 14G | 23% / 0% |

We conduct runtime tests on a single L20 GPU with 48GB VRAM. For each method, we use the same pairwise LoRA combinations as in Table 5 and Table 6, with identical prompt and image resolution, generating 100 results per combination and averaging the runtime. The results show that, despite our method being implemented on the DiT model with significantly more parameters than the U-Net, its runtime and memory usage remain highly competitive: it introduces only about 20% more time overhead compared to our baseline, whereas CLoRA and OMG incur over 400% additional runtime relative to their respective baselines. This further demonstrates the efficiency of our mask computation strategy.                                                              NEW

## G    FAILURE CASE ANALYSIS

Here, we analyze the failure cases and discuss possible solutions. Most failures arise as the number of target subjects increases, making scene understanding increasingly difficult for the base model and leading to hallucinations. As shown in Fig. 10, the model incorrectly concentrates the attention of two characters activation tokens on the same region, while another character exploits the gap and occupies the right side. This ultimately results in Character 2 disappearing and Character 3 appearing twice. We believe this occurs because the base models scene comprehension has already deteriorated under such complex settings.                                                                              NEW

A straightforward solutionsimilar to Mix-of-Showis to let users manually specify fixed spatial regions to constrain where each LoRA can take effect, or to introduce an additional trained module to predict where each character should appear in latent space. However, the goal of FreeFuse is precisely to free users from such burdensome inputs and the need for extra models or parameters. Other alternatives include iteratively determining the effective region for each LoRA and recomputing activation-token attention weights for other subjects after each assignment, but these approaches also introduce significant complexity. We believe that as more powerful DiT models emerge and their scene-understanding capability improves, this issue will naturally diminish.                                              NEW

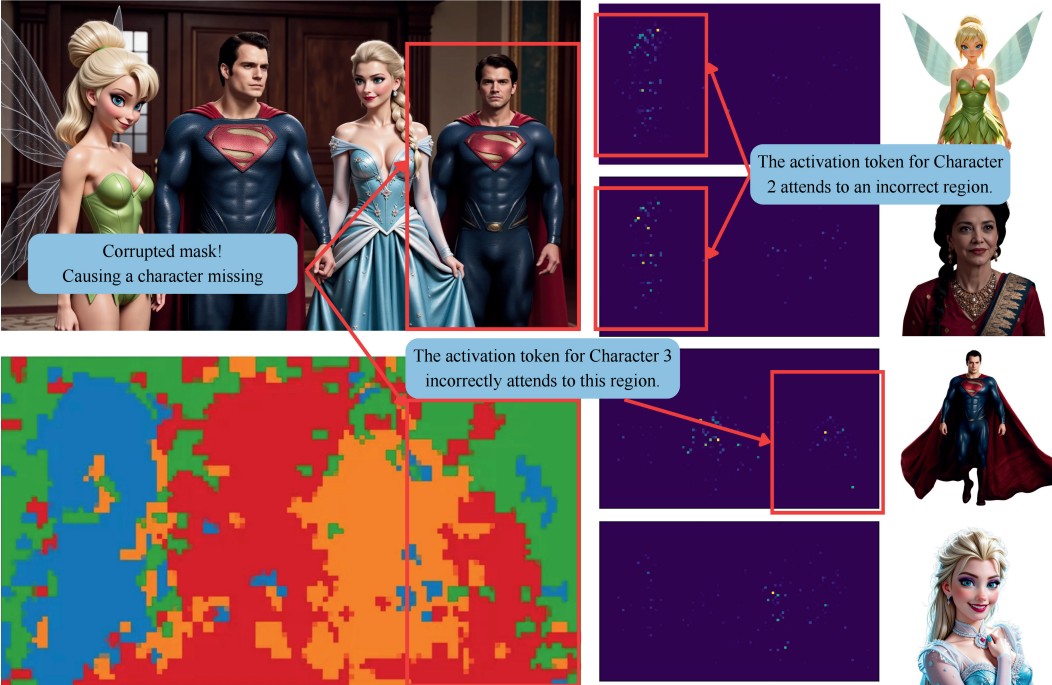

Figure 10: The failure cases mainly emerge as the number of LoRAs increases, making it progressively harder for the base model to understand the scene. Eventually, multiple characters begin to focus on the same region. For example, in this case, Characters 1 and 2 almost entirely attend to the same area, indicating that the model has hallucinated their spatial positions. Character 3 then exploits this gap and takes over the rightmost region, resulting in Character 3 appearing twice and Character 2 disappearing in the final output.                                                                          NEW

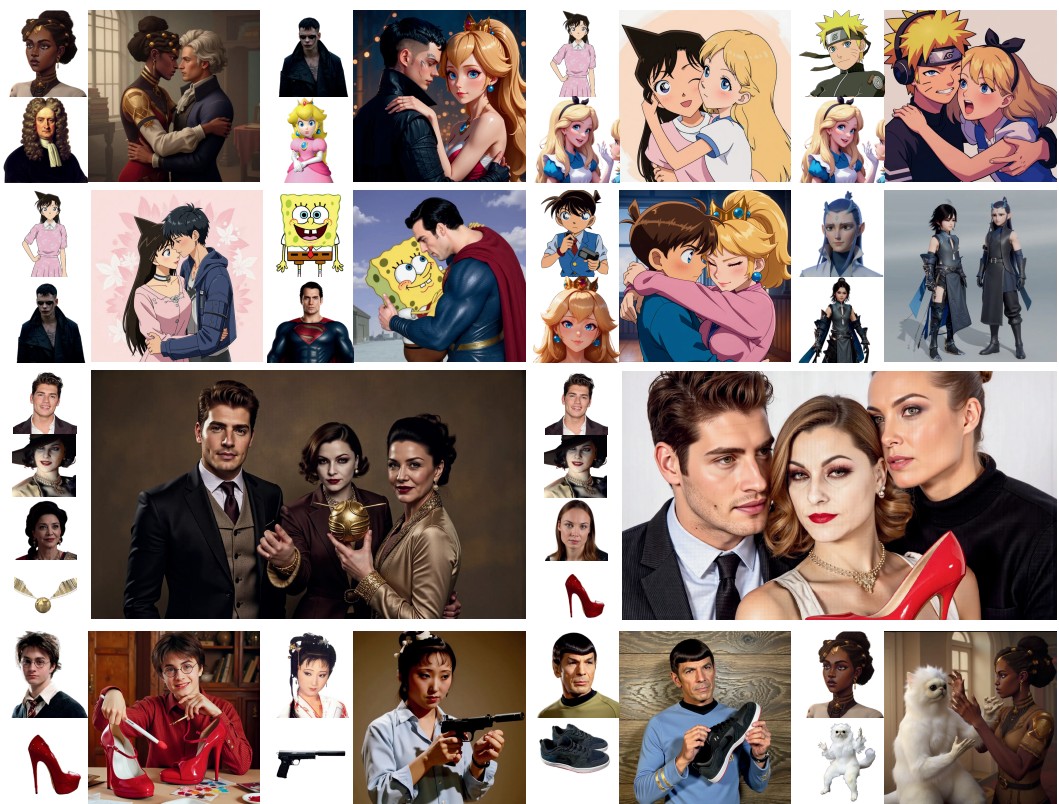

Figure 11: Our method performs exceptionally well in multi-character and object fusion, in blending cartoon and photorealistic characters, and in integrating characters with objects.

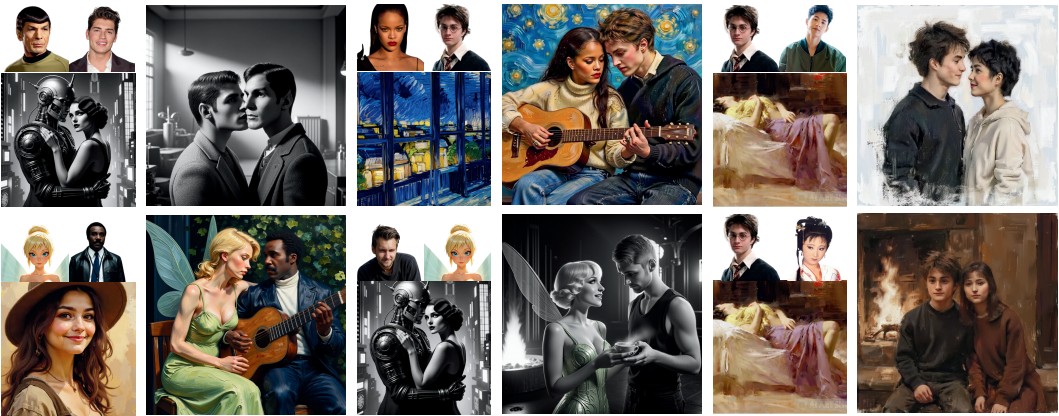

Figure 12: By simply applying an all-ones mask to style LoRAs, our method also works effectively in the presence of style LoRAs.

# H MORE RESULTS

## H.1 MORE QUALITATIVE RESULTS

Here, we showcase several applications of our method on community LoRAs, including character-character fusion, characterobject fusion, and cooperation with style LoRAs in Fig 11 and Fig 12. Our approach supports joint generation with up to four LoRAs, demonstrating strong performance and robust generalization across diverse application scenarios.

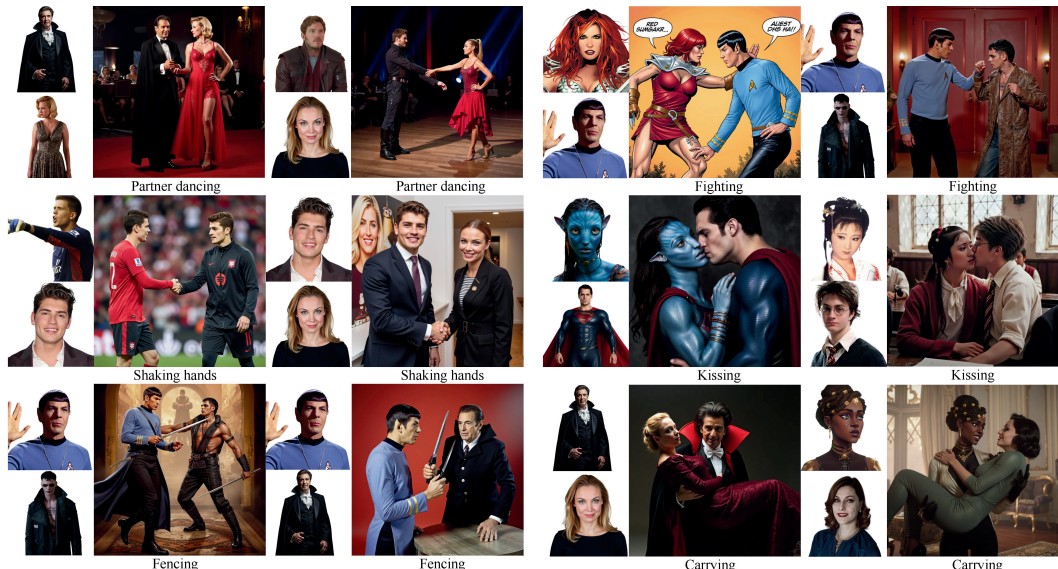

Figure 13: Additional action-level results further demonstrate that our method can generate complex interaction scenarios without requiring any extra components or special techniques.

## H.2 MORE QUALITATIVE RESULTS ON COMPLEX INTERACTIONS

To further demonstrate the effectiveness of our method in complex human interactions, we present additional visualizations on more challenging interaction patterns. We evaluate other difficult actions introduced in DreamRelation, namely partner dancing, fighting, shaking hands, kissing, fencing, and carrying. The visualizations in Fig. 13 show that our method can generate these complex interactions without any special design, whereas DreamRelation requires training a separate action-specific LoRA for each interaction type.                                                                                          NEW

## H.3 FLEXIBLE STYLE TRANSFER

We observe that when combining manga-style and photorealistic LoRAs, the base model naturally tends to produce images with a unified style. We speculate that this is because most training images seen by the base model exhibit consistent stylistic coherence, which also aligns with general human preference. In contrast, Mix-of-Show and Orthogonal Adaptation rely on Regional Control Sampling, which prevents them from producing style-consistent outputs and often results in cut-and-pastelike inconsistencies. Our method, however, allows users to flexibly control the style through prompts. As shown in Fig. 14, if users are not satisfied with style-unified outputs, they can use prompting to enforce each characters original style.                                                                NEW

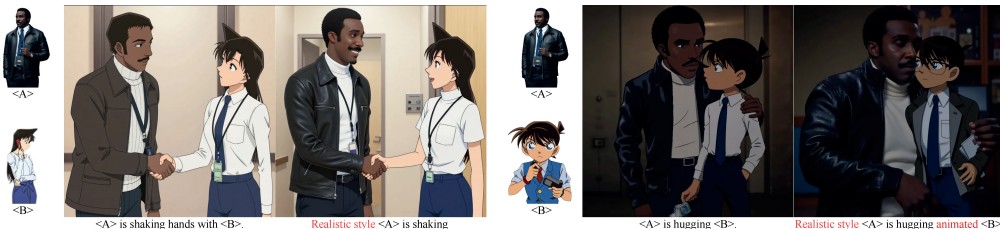

Figure 14: With simple prompt cues, users can freely switch between rendering all characters in a unified style (e.g., all in a manga style) or preserving each characters original style.

## H.4 MORE QUALITATIVE COMPARISONS

1026
1027
1028
1029
1030
1031
1032
1033
1034
1035
1036
1037
1038
1039
1040
1041
1042
1043
1044
1045
1046
1047
1048
1049
1050
1051
1052
1053
1054
1055
1056
1057
1058
1059
1060
1061
1062
1063
1064
1065
1066
1067
1068
1069
1070
1071
1072
1073
1074
1075
1076
1077
1078
1079

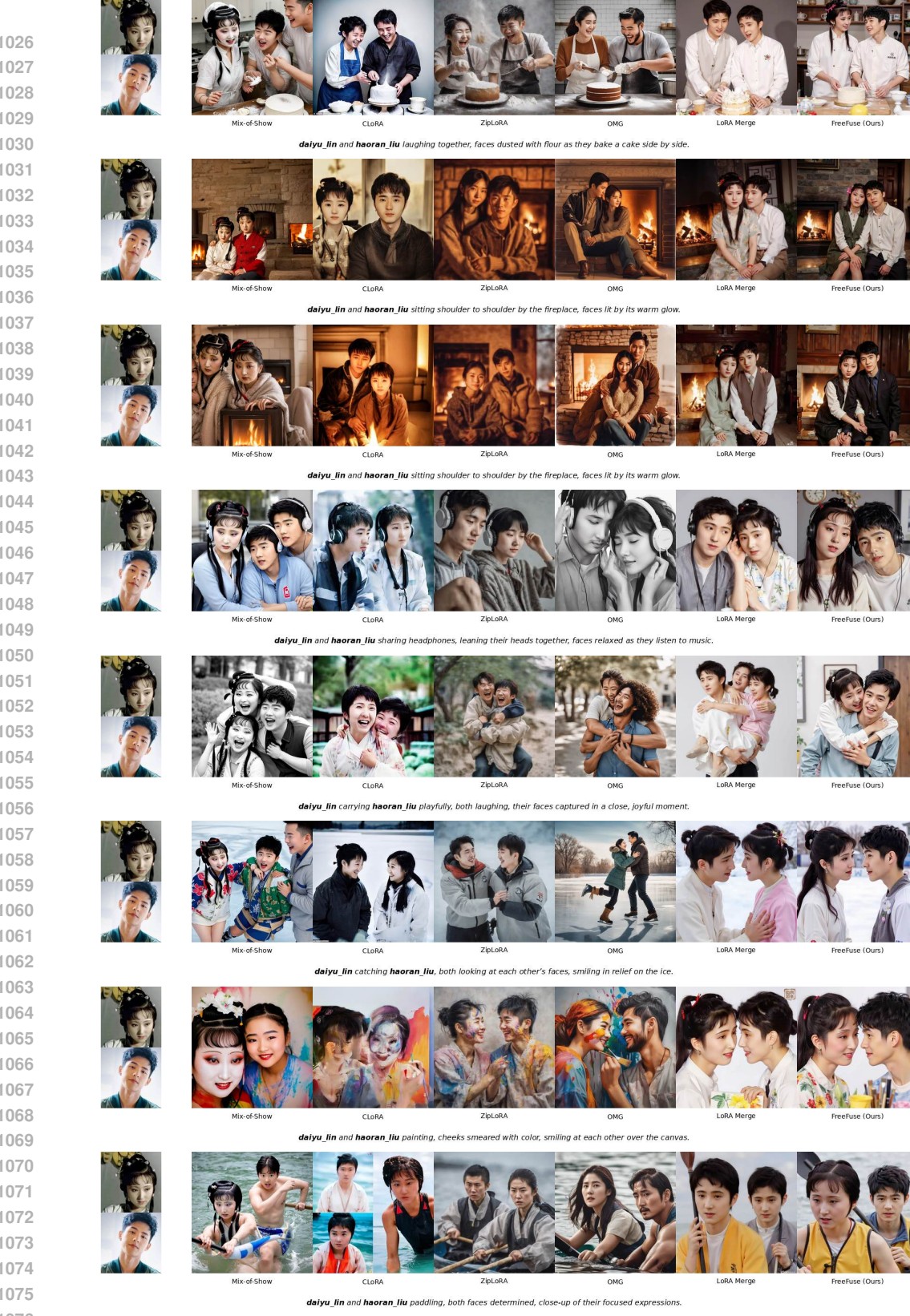

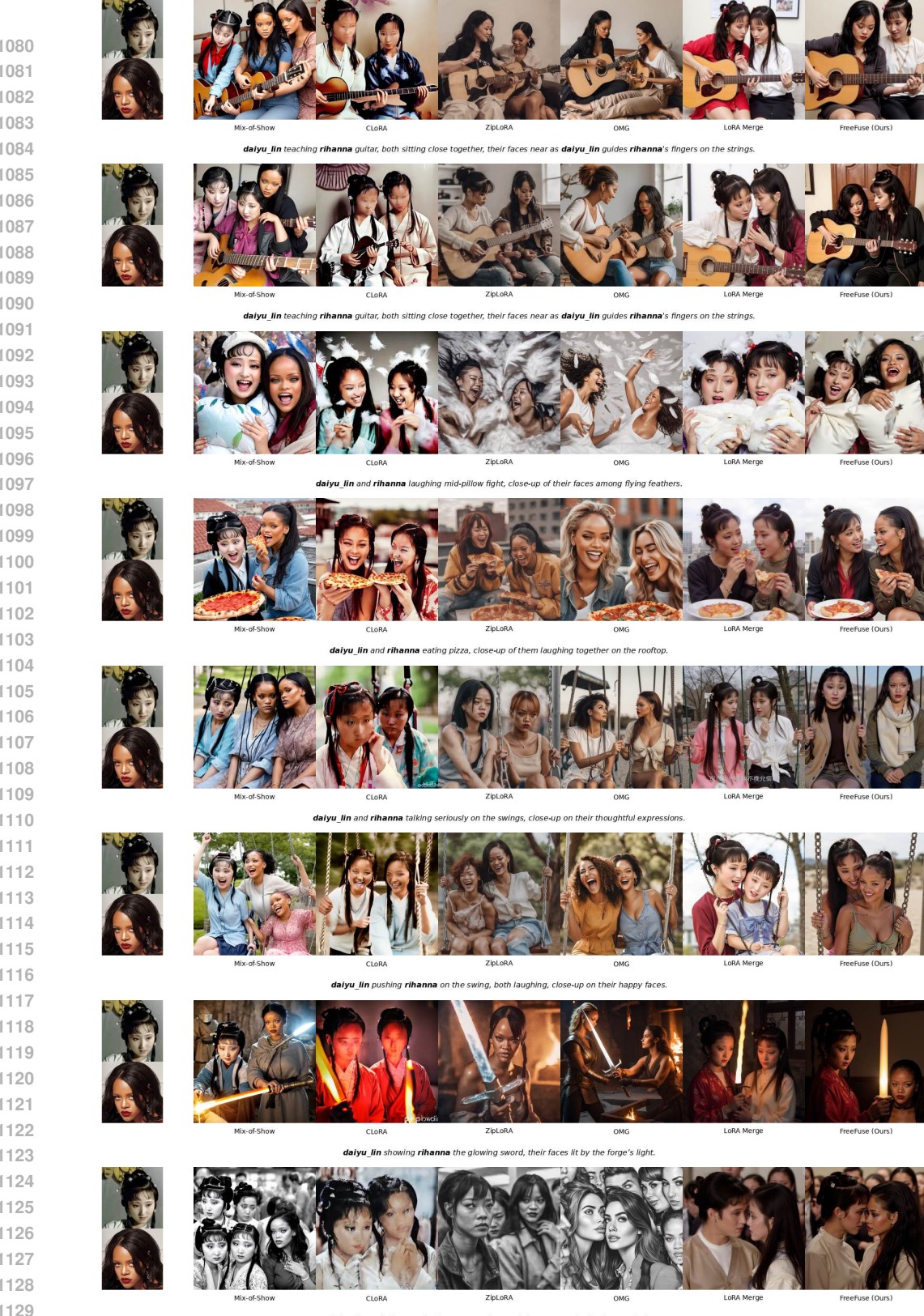

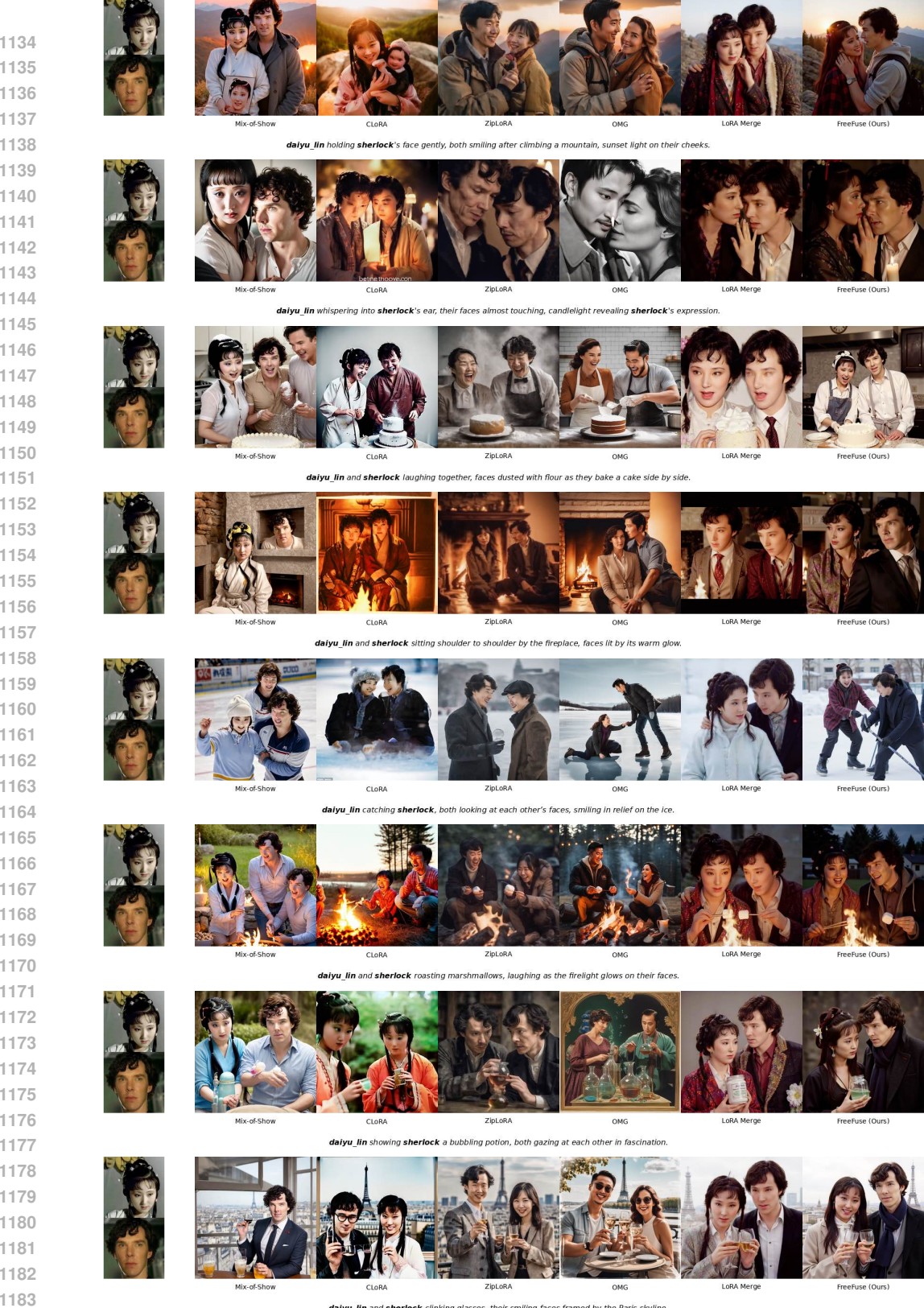

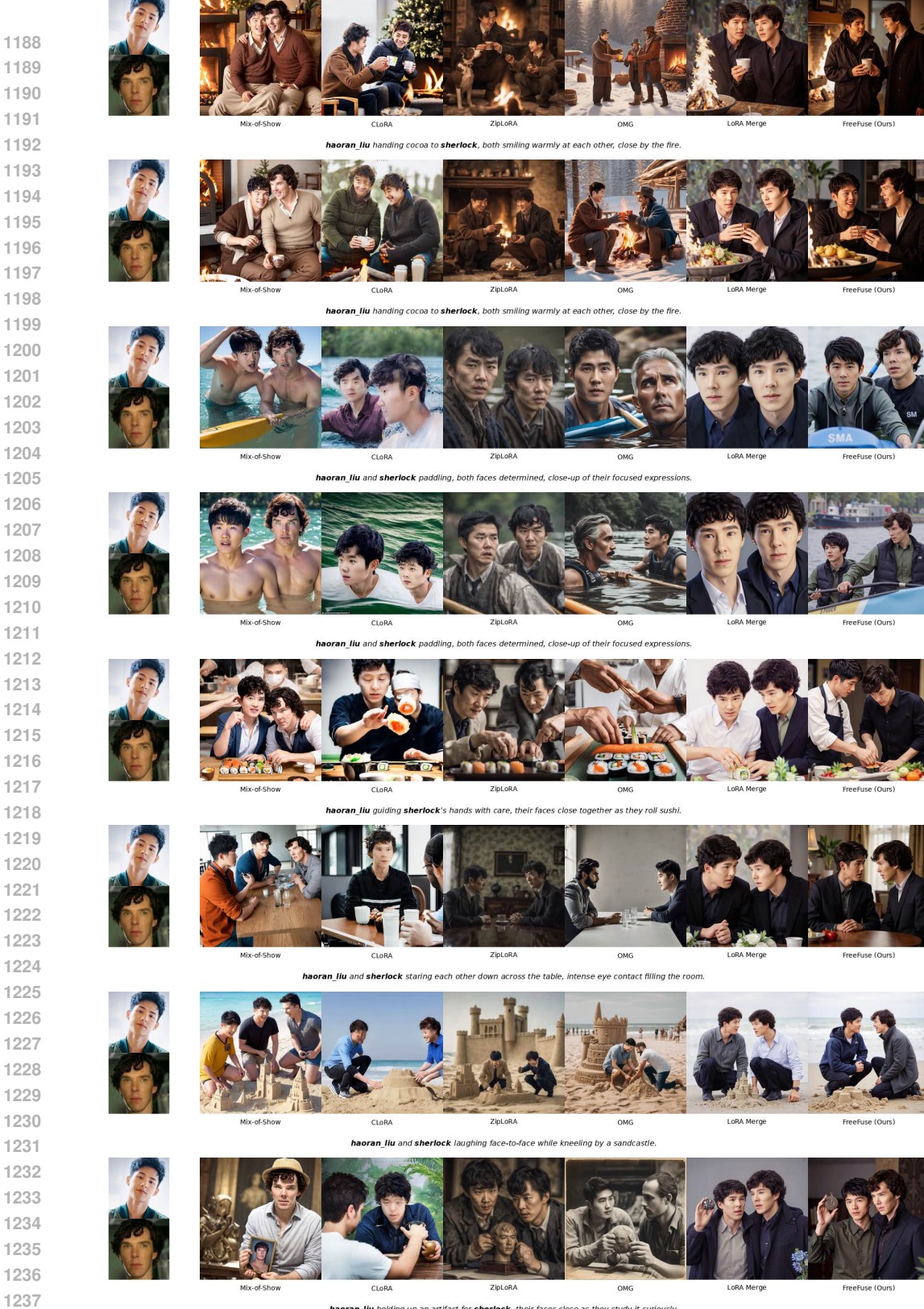

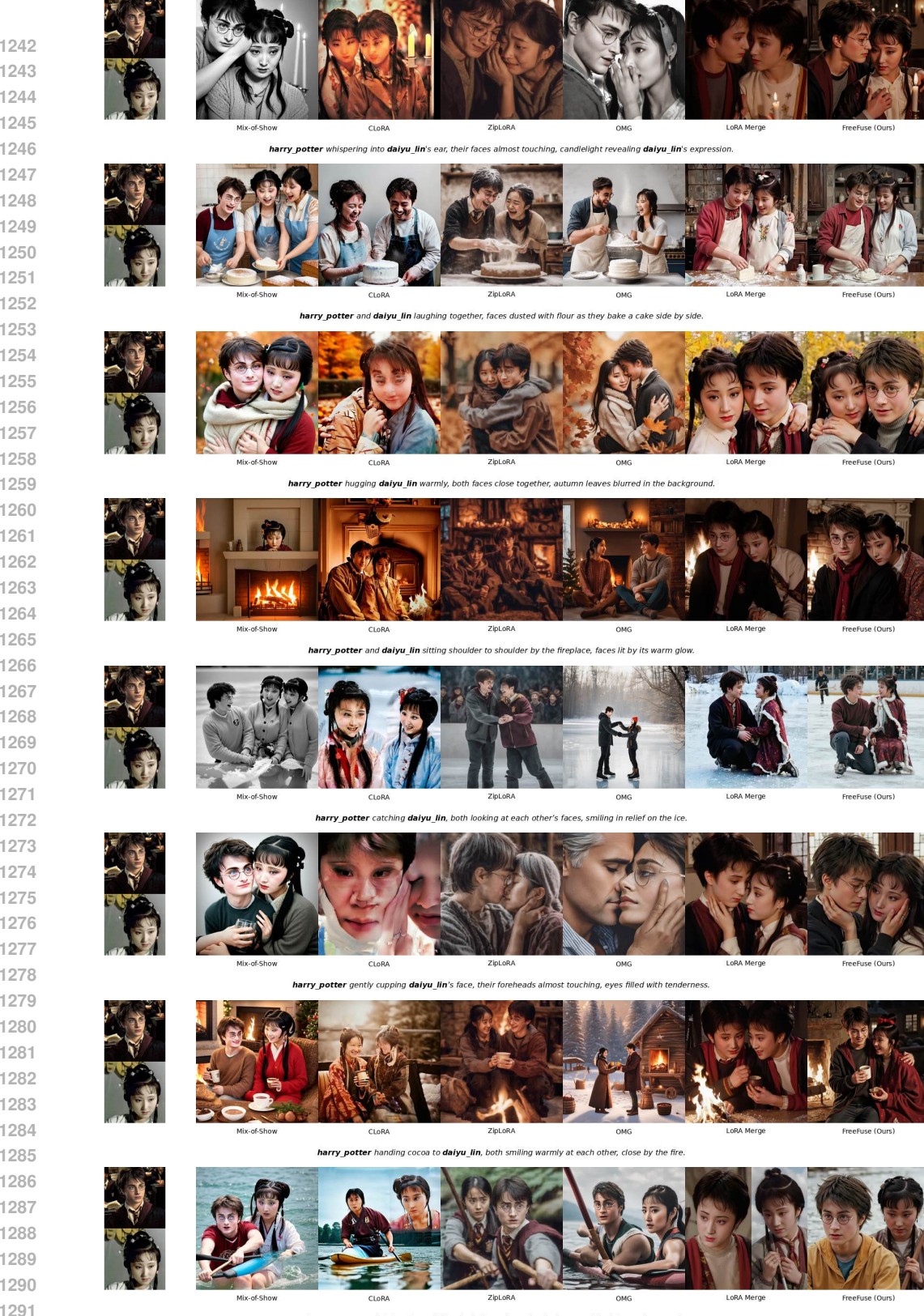

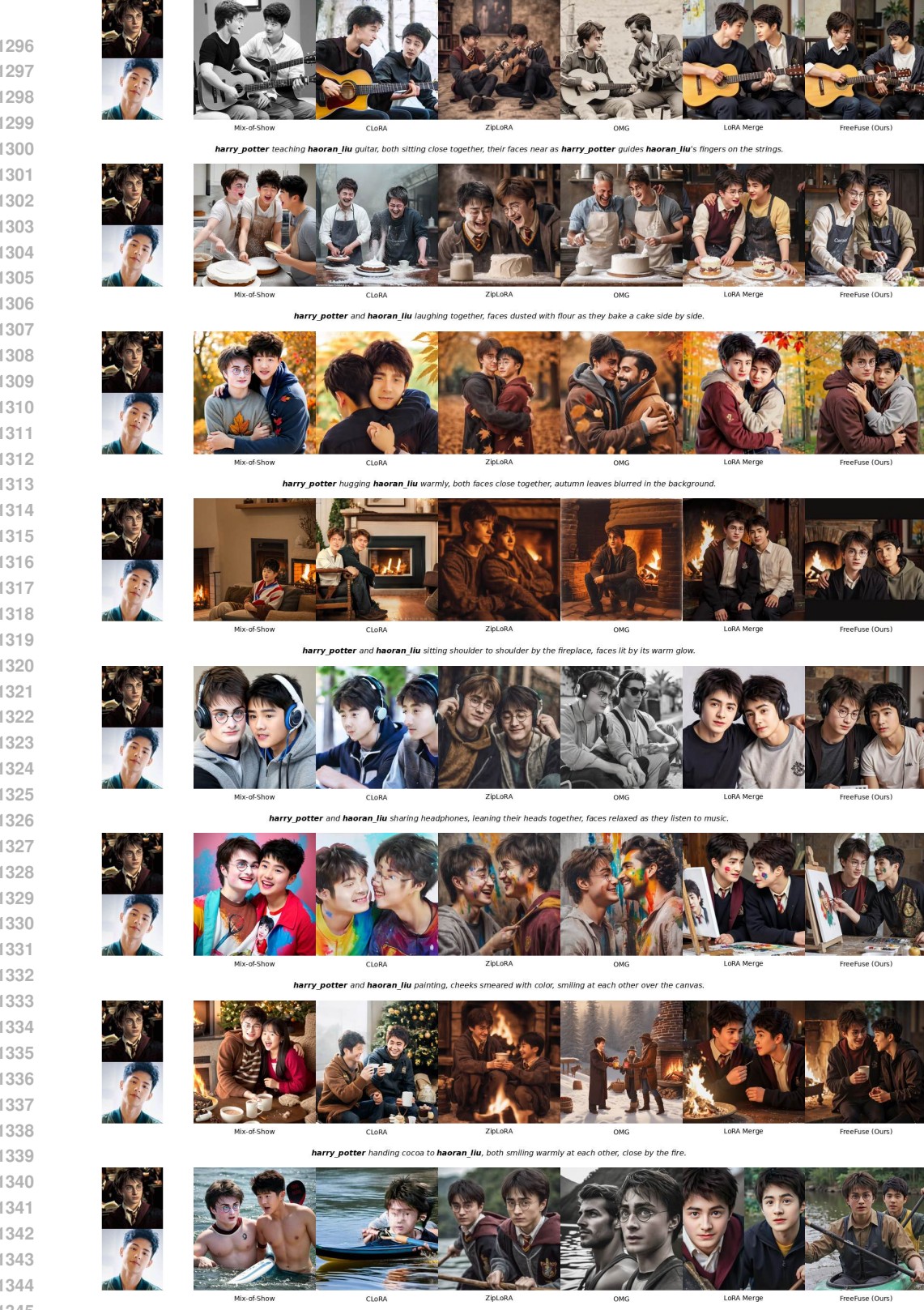

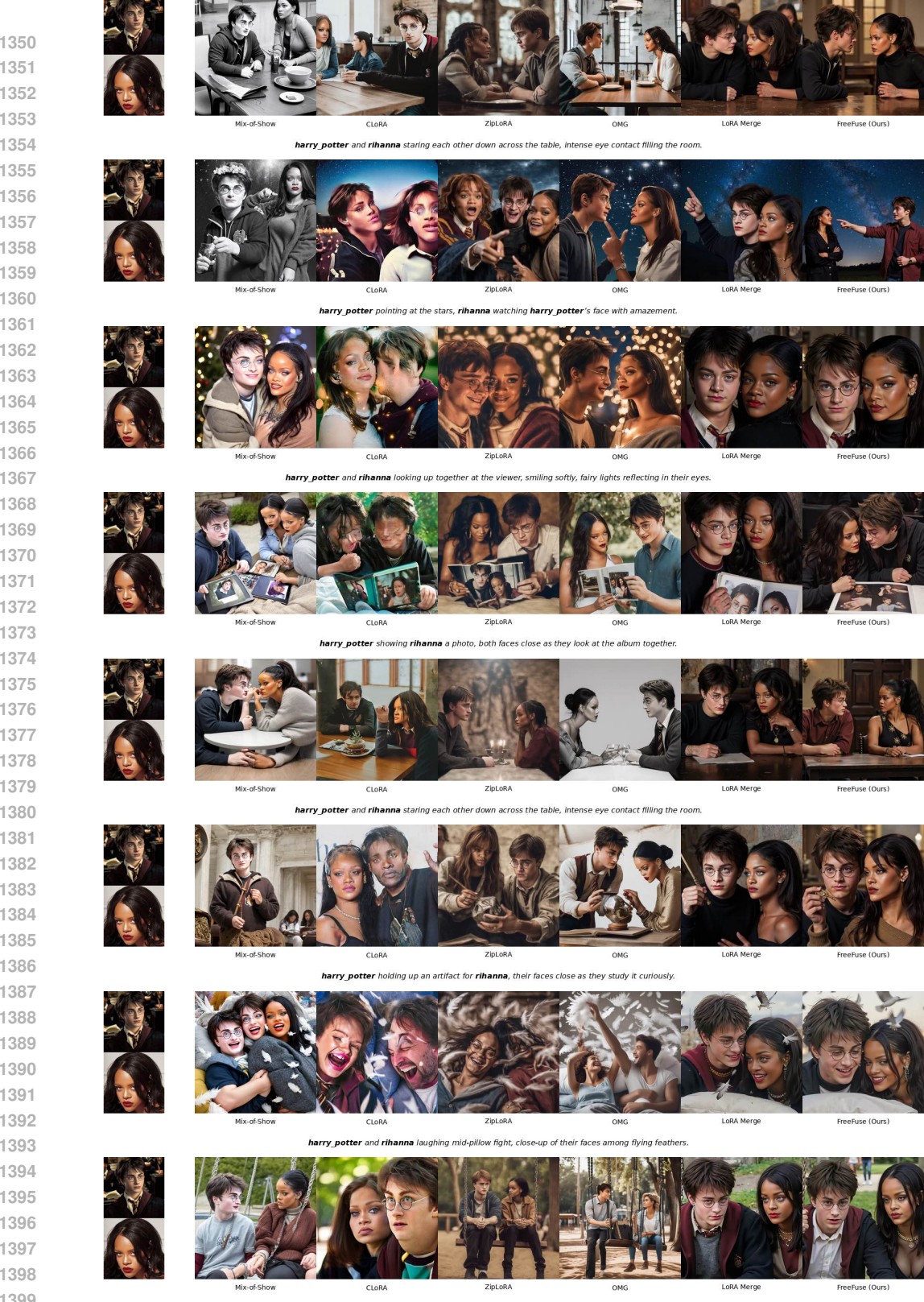

