# OpenReview forum: "FreeFuse: Multi-Subject LoRA Fusion via Auto Masking at Test Time"
_ICLR.cc/2026/Conference — Submitted to ICLR 2026_

### Official Review · Reviewer_hgmv · 2025-10-18

**Soundness:** 2
**Presentation:** 2
**Contribution:** 2
**Rating:** 4
**Confidence:** 4

**Summary:**

The paper presents a method for personalized multi-subject generation via LoRA fusion. Given several LoRAs, each trained on a specific subject separately, the goal is to generate a single image containing multiple personalized subjects.
The proposed method consists of two main stages. In the first stage, a mask is generated for each subject, indicating the image regions where the corresponding LoRA should be applied. In the second stage, these masks are used to fuse the predictions from the different LoRAs by multiplying each LoRA's output with its corresponding mask.

Merging multiple LoRAs using masks is a well-known technique in the community (https://github.com/lifeisboringsoprogramming/sd-webui-lora-masks?tab=readme-ov-file). Therefore, the main contribution of the paper lies in its automatic mask generation approach, which requires no user input or external model.

The method is evaluated both qualitatively and quantitatively against several approaches for personalized multi-subject generation.

**Strengths:**

- The results presented in the paper are of high quality, demonstrating good identity preservation and in most cases plausible prompt adherence.
- The paper provides relevant background on competing methods and clearly differentiates itself from the baselines.
- A large gallery of qualitative results is presented, effectively showing the method's performance.
- The paper includes multiple quantitative metrics and shows consistent advantages across all of them.

**Weaknesses:**

- Section 3.1 lacks clarity in both reasoning and presentation:
    - If the goal of this subsection is to demonstrate that Equation 4 holds, it would be more direct to compute each term in the equation across multiple images, timesteps, and layers, and then show their similarity. The discussion about the queries, keys, values, and feed-forward layers seems unnecessary for this purpose.
    - In Figure 4, the average MSE loss when disabling to_q and to_k appears comparable to the loss when disabling to_v. While the qualitative example shows a difference, this is based on a single sample. Therefore, I am not fully convinced by the conclusion supporting Equation 2.
    - Moreover, it is unclear why, if the LoRA outputs are typically 1-2 orders of magnitude smaller than the base model's outputs, they would affect the Q,K layers differently from the V,FF layers. This further raises doubts about the validity of Equation 2.
    - The computation in Figure 3a is not entirely clear.

- Given that the method is easy to implement (this is an advantage), it would strengthen the paper to include experiments on additional base models (e.g., SD, SDXL) for a fairer comparison with existing baselines.

- Since the main contribution lies in the mask extraction, this part should be evaluated more thoroughly and compared with other techniques. For instance, how does it perform relative to a simple average of attention maps across timesteps and layers, where the queries correspond to image pixels and the keys correspond to subject-related prompt tokens (as in Prompt-to-Prompt)? There are also related approaches such as Readout-Guidance and Self-Guidance that could serve as baselines for mask extraction.

- The paper only demonstrates results for two subjects, whereas other works (e.g., Mix-of-Show) show examples with more subjects.

- In some examples, prompt adherence appears weaker than in competing methods. For example, on page 15, in the top example, the faces are not dusted with flour, and in the 7th row, the faces are not smeared with color as specified in the prompt.

**Questions:**

See the questions in the Weaknesses Section.

---

> ### Author Response · Authors · 2025-11-24
> **Rebuttal from Authors of Paper8708 to Reviewer hgmv**
>
> We sincerely thank reviewer for the careful reading of our method section. We address the concerns as follows:
>
> **Regarding W1(Clarity in both reasoning and presentation).**
>
>  Our goal is not to provide a rigorous mathematical proof, but rather to offer an intuitive analysis demonstrating that applying masks effectively mitigates feature conflicts between LoRAs. We have revised our analysis. In the task of generating $N$ subjects with $N$ LoRAs applied jointly, direct joint inference introduces two forms of LoRA interference. First, in the Attention-V and FF layers, for the region $R_i$ corresponding to subject i, all LoRA outputs are summed, causing direct conflicts. Second, each LoRA’s output may be aggregated into the tokens within $R_i$ through attention, further introducing cross-LoRA interference.
>
>  To address this, we first apply masks to ensure that only LoRA i contributes within region $R_i$ in the FF and V pathways. Then, leveraging the well-documented locality of attention, we argue that tokens inside $R_i$ primarily attend to themselves, so the aggregation step does not bring substantial features from other LoRAs into $R_i$.
>  Consequently, our method remains simple while effectively mitigating cross-LoRA feature conflicts.
> Please refer to the updated **section “Masking LoRA Outputs for Effective Subject Feature Preservation.”**
>
> **Regarding W2(Transfering to other base model).**
>
>  Our method aims to unlock the inherent multi-LoRA, multi-subject generation capabilities of DiT-based models. Therefore, we additionally transfer our method to Stable Diffusion 3.5, which also adopts a DiT architecture. Results in **Appendix E** demonstrate that the method remains highly effective in this setting.
>
> **Regarding W3(Comparing to other possible mask extracting methods).**
>
>  We fully understand the reviewer’s desire for a thorough comparison against alternative mask-extraction strategies. Our ablation includes multiple settings.
>  For timestep selection, we observe that DiT models typically determine global composition in the first 4–7 steps, making these steps the optimal choice for mask extraction.
>  For layer selection, only the final double-stream blocks produce stable, structurally meaningful cross-attention maps; earlier blocks often do not. Hence, operations such as *“simple averaging of attention maps across timesteps and layers (pixel queries × subject-token keys)”* not only incur significant computational overhead but also reduce mask accuracy. See **Figure 8b**.
>
> We also compare hard and soft masking strategies inspired by Self-Guidance and Readout Guidance.
> - Hard masks either collapse to selecting only sink tokens or produce severe overlap due to inaccurate attention maps, and lack block-level voting, leading to masks with holes.
> - Soft masks are dominated by sink tokens, causing nearly all other tokens to have weights around ~0.4, which effectively degenerates into mixing LoRA outputs at a fixed 0.4 ratio.
>
> See **Figure 8a** for details.
>
> **Regarding W4(Scalibilty Demonstration).**
>
>  We further present results on multi-subject generation. Please refer to **Figure 6**, **Figure 11**, and **Table 3**, which collectively demonstrate strong multi-subject capability.
>
> **Regarding W5(Some cases do not strictly follow the prompt).**
>
>  For the occasional failure case mentioned by the reviewer, this issue is not introduced by FreeFuse, but rather stems from the more pronounced overfitting observed in DiT-based LoRA training. We examined the character LoRAs trained on Flux. After convergence, modifying facial attributes via prompts (e.g., smeared with color) becomes difficult, suggesting that Flux-trained LoRAs may overfit strongly to character facial features, making such attributes harder to alter.

---

### Official Review · Reviewer_nDjA · 2025-10-22

**Soundness:** 2
**Presentation:** 2
**Contribution:** 2
**Rating:** 2
**Confidence:** 5

**Summary:**

This paper addresses the critical challenge of feature conflicts in multi-subject text-to-image generation when fusing multiple subject LoRAs during joint inference. Existing methods often require retraining, auxiliary segmentation models, user-defined prompts/regions, or pre-inference LoRA weight merging—limiting their practicality. To solve this, the authors propose FreeFuse, a training-free framework tailored for Diffusion Transformers (DiTs) that enables seamless multi-LoRA fusion via automatically derived subject masks.
FreeFuse requires no additional training, no modifications to existing LoRAs, no auxiliary models, and only needs users to provide LoRA activation words. Experiments on FLUX.1-dev show it outperforms baselines (LoRA Merge, ZipLoRA, OMG, Mix-of-Show, CLoRA) across metrics: e.g., achieving a VLM score of 74.03 (vs. 57.74 for Mix-of-Show) and a 10-Pass LVFace score of 0.4685 (vs. 0.4417 for Mix-of-Show). It also excels in complex subject interactions (e.g., hugging, whispering) that prior methods struggle with.

**Strengths:**

- Unlike methods requiring retraining (Mix-of-Show) or auxiliary segmentation models (OMG), FreeFuse operates entirely at test time. It needs no LoRA modifications, no user-defined region prompts, and only requires LoRA activation words—enabling seamless integration into standard text-to-image workflows.
- FreeFuse addresses key flaws of attention-based mask extraction (e.g., attention sink, noisy pixel-wise maps) via heuristic filtering, self-attention locality exploitation, and superpixel voting. This ensures masks are accurate and spatially coherent without human intervention.
- Most existing multi-LoRA fusion methods (e.g., ZipLoRA, CLoRA) are designed for UNet-based models. FreeFuse targets DiTs (e.g., FLUX.1-dev), filling a critical gap in supporting state-of-the-art transformer-based diffusion models.
- Experiments use diverse metrics to evaluate identity preservation (LVFace), feature similarity (DINOv3), human preference (DreamSim, HPSv3), and prompt adherence (VLM). It compares against 5 strong baselines and validates on complex interaction scenarios—strengthening the credibility of its effectiveness.

**Weaknesses:**

- FreeFuse’s core application scenario (generating multi-subject interaction images, e.g., hugging, face-to-face talking) overlaps heavily with methods designed for character relationship synthesis (e.g., DreamRelation). However, the paper fails to cite or compare with such works, leaving its novelty relative to state-of-the-art interaction-focused generation methods unclear.
- All experiments rely on a small, fixed set of subjects (e.g., daiyu_lin, haoran_liu, Harry Potter, Rihanna). There is no validation on more diverse identities—such as subjects of different ethnicities, ages, artistic styles (e.g., cartoon vs. photorealistic), or non-human subjects (e.g., animals, fictional creatures). This limits the demonstration of FreeFuse’s generalizability.
- The paper excludes recent training-free multi-LoRA fusion methods beyond K-LoRA (e.g., latest variants of LoRA merging or dynamic gating approaches). This incomplete comparison may overstate FreeFuse’s advantages by ignoring competing methods with similar practicality.
- The authors explicitly acknowledge that FreeFuse degrades when the number of subject LoRAs increases. As each LoRA’s masked region shrinks, features from other LoRAs are more likely to intrude—making the method ineffective for scenarios with 5+ subjects.
- In scenes with heavy subject overlap (e.g., two people embracing with intertwined limbs), the attention-based masks may fail to accurately separate individual subjects. This leads to residual feature conflicts that FreeFuse cannot resolve.

**Questions:**

- Why were methods for character relationship synthesis (e.g., DreamRelation) not compared? How does FreeFuse’s performance on multi-subject interaction tasks differ from these methods, especially in terms of interaction naturalness and identity preservation?
- What is the reason for using only a small set of fixed identities in experiments? If tested on more diverse subjects (e.g., elderly individuals, non-Western ethnicities, cartoon characters), would FreeFuse’s metrics (e.g., LVFace similarity, mask accuracy) remain stable, or would performance degrade?
- The paper notes performance issues with many LoRAs, but no potential solutions are proposed. Could dynamic mask resizing, multi-scale attention fusion, or adaptive LoRA activation weights address this limitation?
- Can FreeFuse be adapted to other DiT models (e.g., Stable Diffusion 3) or UNet-based models? If so, would the mask generation step (e.g., layer selection, denoising step) require significant adjustments?
- How would FreeFuse handle scenes with extreme subject overlap (e.g., a group hug with 3+ people)? Is there a strategy to improve mask accuracy in such cases, such as integrating lightweight geometric cues?

---

> ### Author Response · Authors · 2025-11-24
> **Rebuttal from Authors of Paper8708 to Reviewer nDjA**
>
> We sincerely thank the reviewer for recognizing the simplicity, usability, and technical advantages of our method. Below we address the raised concerns point by point:
>
> **Regarding W1 & Q1(Comparison to DreamRelation).**
>
>  Our work focuses on analyzing and resolving feature conflicts that arise when jointly sampling multiple subject LoRAs. In contrast, methods such as DreamRelation rely on SSR Encoders or IP-adapter encoders, where subject information is provided via prompt images rather than LoRAs. This setting differs fundamentally from our research scope. Nevertheless, we acknowledge DreamRelation as an important representative of fine-grained control. However, it requires separate LoRAs for different actions, user-provided regional prompts, and an additional image-prompt encoder—making the pipeline complex and heavily skill-dependent.
>
>  Our method aims to free users from such burdens: apart from supplying activation words for each LoRA, no extra inputs are needed. Despite this minimal interface, our approach performs well even for complex actions. **Table 1** in the main paper provides a detailed comparison.
>
> **Regarding W2 & Q2(Reason for experimenting on 5 real character LoRAs and generalization beyond human characters).**
>
>  As stated in the initial submission, *“Conflicts between character LoRAs are often the most severe and can effectively reflect each method’s actual capability in mitigating inter-LoRA feature conflicts.”*
>  Therefore, we intentionally focus on human-character LoRAs to emphasize performance on the hardest cases. To further demonstrate generality across LoRA types, we additionally evaluate on 35 character LoRAs (cartoon, 3D, and anime) and 8 object LoRAs. Results show that our method consistently produces strong compositions across all categories. Please refer to the updated **experimental** section and **Figure 6**, **Figure 11**, **Table 3**.
>
> **Regarding Q3 & W3(Latest LoRA merging and gating methods).**
>
>  We have included Orthogonal Adaptation (CVPR 2025) as the latest LoRA-merging baseline. Our method retains a clear advantage over this approach as well.
>
> **Regarding W4(Scalibility on more LoRAs).**
>
>  We fully understand the reviewer’s concern about scalability. To address this, we added experiments involving multiple characters plus objects. Our method remains effective even when 4 LoRAs are activated simultaneously, without requiring users to manually specify spatial regions for each LoRA. In contrast, prior methods such as Mix-of-Show require such explicit region control to generate multiple characters. Please see **Table 3**, **Figure 6**, and Appendix **Figure 11**. As the number of LoRAs continues to increase, performance degradation appears. **Appendix G** provides a detailed analysis: cross-attention maps show that the DiT model starts hallucinating under overly complex scenes, incorrectly placing different characters in the same location, which causes duplication or disappearance. We also discuss potential remedies, including user-provided regional prompts, training a lightweight position-correction module, or recursively determining character placements.
>
> **Regarding Q4(Transfering to other base model).**
>
>  We appreciate the reviewer’s concern about the transferability of our approach. Our goal is to unlock the DiT model’s ability to generate multi-character scenes from multiple LoRAs without external modules. Experiments on SD3.5-Large confirm that our method transfers well without major modifications. The only change is the mask extraction location: Block 17 (double-stream) in Flux vs. Block 25 (Transformer) in SD3.5. Please refer to **Appendix.E** for more details.
>
> **Regarding Q5 & W5(Scenes with heavy subject overlap).**
>
>  Our paper already includes numerous cases with strong inter-subject overlap (e.g., *two people embracing with intertwined limbs*), demonstrating that our method handles such scenarios effectively. Please refer to the teaser, **Figure 6**, and **Figure 11** for examples of characters hugging and interacting closely.

---

> ### Comment · Reviewer_nDjA · 2025-11-26
>
> Thank you for the author's reply. Although the new results tested more examples, the results in Figures 6, 11, and 12 show that ID preservation is not good enough for a wider range of characters and styles. Furthermore, despite claiming to handle many complex interactions, the results only demonstrate relatively simple interactions. Can the method presented in this paper support arbitrary interactions? Or is there a fixed pool of interactions? If arbitrary interactions are possible, what mechanism maintains this stability? Besides, although DreamRelation is fundamentally different in its implementation path, it can achieve a lot of character interactions in relationships.

---

> ### Author Response · Authors · 2025-11-27
> **Rebuttal from Authors of Paper8708 to Reviewer nDjA**
>
> Thank you for the reviewer’s timely response.
>
> **Regarding the capability for complex interactions.**
>
>  **Our method supports arbitrary actions in arbitrary scenes**. In our previous demonstrations, we primarily showcased the hugging interaction because, in your earlier review, you specifically emphasized the need to observe scenarios with “heavy subject overlap.” To further illustrate the capability of our method, **Figure 13** presents six additional interactions from DreamRelation: **partner dancing, fighting, shaking hands, kissing, fencing, and carrying**, beyond the hugging example already provided. These results show that our method can naturally handle a wide variety of complex multi-character interactions without any special, interaction-specific design.
>
> The key reason behind this strong capability lies in how the influence regions of multiple LoRAs are determined: they are entirely guided by the attention maps of the base model. This allows our method to preserve the base model’s inherent understanding of both the prompt and the scene to the greatest extent. In contrast, previous approaches rely on **regional sampling**, which tries to force each subject to appear at a fixed spatial location. Such hard constraints significantly limit the expressive capacity of the base model and prevent it from fully leveraging its built-in scene comprehension ability.
>
>
> ---
>
> **Regarding the concern that “the results in Figures 6, 11, and 12 show that ID preservation is not good enough for a wider range of characters and styles.”**
>
> We observed that when anime-style characters and real-human characters are generated together, diffusion models tend to produce results in a unified visual style (e.g., both rendered in a realistic style, or both in an anime style). This behavior aligns with the data distribution of the model’s pretraining corpus, where cross-style character interactions (e.g., anime + real human) are relatively rare in real-world datasets. Moreover, such unified-style outputs are often closer to typical user preferences.
>
> If users explicitly require each character to maintain its own original style, this can be achieved through simple prompt engineering, one only needs to clearly specify the intended style for each character in the prompt, as shown in **Figure 14**.
>
> As for the decrease in ID preservation after applying Style LoRAs, this issue is often caused by bias inherent in the Style LoRAs themselves. For example, a LoRA trained on the painting style of Pino Daeni may inadvertently push characters toward the characteristics commonly found in his artworks. Because Pino Daeni predominantly painted white subjects, the corresponding Style LoRA may attempt not only to transfer the style but also to alter the ethnicity of characters, thereby reducing ID preservation.

---

### Official Review · Reviewer_xeE2 · 2025-11-01

**Soundness:** 3
**Presentation:** 3
**Contribution:** 3
**Rating:** 6
**Confidence:** 4

**Summary:**

This paper presents FreeFuse, a training-free and segmentation-free framework for multi-subject text-to-image generation by fusing multiple subject LoRAs directly at inference. Instead of retraining or merging LoRAs, FreeFuse derives context-aware subject masks from cross-attention maps and applies them to LoRA outputs, mitigating feature conflicts during joint inference. The key insight is that context-aware dynamic subject masks can be automatically derived from cross-attention layer weights, which well approximate the case where each LoRA is integrated into the diffusion model and used individually. FreeFuse extracts these masks from a single attention block and denoising step, achieving efficiency advantages over prior methods such as CLoRA, OMG, and Mix-of-Show. FreeFuse outperforms several baselines on DINOv3, DreamSim, HPSv3, and Gemini-2.5 VLM metrics, with notably higher VLM score (74.03 vs 57.74 for Mix-of-Show).

**Strengths:**

1. Clear problem motivation and theoretical grounding: This paper identifies intense competition among LoRAs in key subject regions as the source of failures in joint inference, supporting it with cosine-similarity visualizations of latent interference.

2. Elegant and efficient formulation: The core mathematical argument formally justifies why masking LoRA outputs approximates isolated inference, showing that locality of attention ensures near-identical representations inside the mask.

3. Attention-based automatic mask extraction: The pipeline introduces attention-sink filtering and superpixel-level voting to ensure spatial coherence. Importantly, it requires no retraining, no LoRA modification, and no external segmentation.

4. Strong empirical evaluation: Evaluation uses five complementary metrics: DINOv3, DreamSim, LVFace, HPSv3, and Gemini-2.5 VLM, covering both perceptual similarity and human preference alignment. FreeFuse achieves the highest VLM score (74.03) and DreamSim 10-pass (0.8052), showing superior realism and consistency.

5. Comprehensive ablation studies: Fig. 7 clearly isolates the effect of attention-sink handling, self-attention maps, and block-level voting; omitting any step causes visible artifacts, reinforcing the necessity of each design.

**Weaknesses:**

1. Lack of runtime and resource benchmarks: Implementation details mention 37s per image on a single L20 GPU, but omit comparisons with CLoRA/OMG or multi-step variants, making it hard to quantify efficiency gains.

2. Potential bias toward photorealistic scenarios. Evaluation prompts emphasize intimate, realistic human interactions. It remains unclear whether FreeFuse generalizes to style LoRAs, cartoons, or abstract concepts.

3. Scalability is unclear with more subjects. Yet no quantitative evidence is provided for how performance degrades beyond two subjects.

**Questions:**

1. Generalization beyond human characters — All evaluation prompts involve human pairs. Have you tested FreeFuse on object+character or style+subject LoRA fusion (e.g., “anime character + van Gogh style”)? If so, do auto-masks still localize meaningfully?

2. Attention-sink filtering parameters — In Eq. 6, you fix p = 1 %. Did you tune p for different resolutions or datasets? How sensitive are final masks and performance to this threshold?

3. Scalability with subject count — Could you report quantitative degradation (e.g., VLM / LVFace scores) for 3-, 4-, and 5-subject scenes to illustrate the scaling limit?

---

> ### Author Response · Authors · 2025-11-24
> **Rebuttal from Authors of Paper8708 to Reviewer xeE2**
>
> We sincerely appreciate the reviewers’ recognition of the usability of our method and the high quality of our results.
>
> **Regarding W1(Lack of runtime and resource benchmarks).**
>
>  We report the full runtime analysis in **Appendix F**. Compared to recomputing masks at every timestep, our approach reduces the overall runtime by ~30%. Despite being implemented on a DiT architecture with substantially more parameters than U-Net, our runtime and memory footprint are comparable to—or even better than—OMG and CLoRA. In contrast, CLoRA incurs over 400% additional overhead relative to its baseline due to frequent attention-map extraction and repeated backward passes.
>
> **Regarding W2 & Q1(Generalization beyond human characters).**
>
>  To further demonstrate the applicability of our method beyond real-human LoRAs, we conducted additional experiments on 35 character LoRAs and 8 object LoRAs spanning cartoon, 3D, anime, and real-human subjects. Results show that our method consistently handles cross-LoRA fusion across all categories; please refer to the Teaser, **Figure 6**, **Figure 11**, and **Table 3**.
>
>  For style LoRAs, integration is straightforward: applying an all-ones mask is sufficient, as style LoRAs induce stylistic shifts rather than feature conflicts. We present extensive style-LoRA results in the Teaser, **Figure 6**, and **Figure 12**.
>
> **Regarding W3&Q3(Scalability with more subjects).**
>
>  We have strengthened the demonstration of our method’s scalability in the revised version. The Teaser, **Figure 6**, and **Figure 11** all showcase compositions involving multiple characters and objects. **Table 3** further quantifies scalability that our method can generalize to four subjects.
>  In **Appendix G**, we analyze failure cases in detail. Cross-attention visualizations suggest that when the scene becomes overly complex, the DiT backbone may hallucinate and place different characters in overlapping regions, causing duplication or disappearance. We discuss several potential solutions, including requiring regional prompts, training lightweight correction models for location prediction, or recursively applying masks. We also believe that as stronger base models with better scene understanding emerge, the upper bound of our method will continue to improve.
>
> **Regarding Q2 (Hyperparameter p).**
>
>  During experiment, we experimented with different values of the pruning ratio p and selected 1% as the optimal value for 1024×1024 resolution (latent size 64×64). A smaller value such as 0.1% (removing at most 4 tokens) fails to reliably eliminate sink tokens, whereas a larger value like 5% (removing up to 200 tokens) may incorrectly prune tokens belonging to the subject, especially when the face is generated near the image boundary.
>  For other resolutions, we recommend scaling p proportionally using
> $0.01\times \sqrt{\frac{H\ \times\ W}{1024\times1024}}.$

---

### Official Review · Reviewer_54Vm · 2025-11-01

**Soundness:** 3
**Presentation:** 3
**Contribution:** 2
**Rating:** 4
**Confidence:** 4

**Summary:**

This paper introduces FreeFuse, a training-free framework for multi-subject text-to-image generation that enables fusing multiple LoRA modules without retraining, external models, or manual prompt engineering. The key insight is that cross-attention maps in diffusion transformer (DiT) models contain sufficient spatial locality to derive subject-specific masks automatically. These masks are then applied to constrain each LoRA’s effect to its relevant region, mitigating feature conflicts during joint inference.
The method is implemented in two stages: (1) mask extraction via filtered attention maps and superpixel-based voting, and (2) mask-guided inference where LoRA outputs are masked at each denoising step. Extensive experiments on FLUX.1-dev demonstrate that FreeFuse outperforms prior works such as OMG, CLoRA, Mix-of-Show, and ZipLoRA across quantitative metrics (DINOv3, DreamSim, LVFace, HPSv3, and VLM scoring). The approach is practical, efficient, and requires no modification to existing LoRA modules

**Strengths:**

The method is entirely training-free, does not modify LoRAs or base diffusion models, and integrates seamlessly with existing workflows. This makes it extremely relevant for real-world adoption. The primary strength of this work lies in its "plug-and-play" nature. By eliminating the need for any additional training, external models, or manual user intervention (like region specifications), FreeFuse presents a highly practical solution to a common problem. Using cross- and self-attention maps for automated subject mask generation is a simple yet powerful concept. Theoretical analysis showing that masked LoRA outputs approximate isolated inference adds credibility.

**Weaknesses:**

1. The most significant weakness is the method's unproven scalability. All qualitative examples in the paper demonstrate a fusion of exactly two subject LoRAs. The authors explicitly concede this limitation in the conclusion, stating that the method's core premise gradually becomes invalid as the number of subject-LoRAs increases. For a paper on multi-subject fusion, the lack of any experimental validation (even as a failure case analysis) for three or more subjects is a major omission.
2. The mathematical justification (Eq. 4) is based on empirical assumptions about attention locality and LoRA perturbation magnitude. While reasonable, it remains an approximation rather than a rigorous derivation.
3. Most baselines (OMG, Mix-of-Show, CLoRA) were implemented on U-Net diffusion models. Evaluating FreeFuse against more recent DiT-native multi-LoRA methods would strengthen the claim of DiT superiority.

**Questions:**

1.Does FreeFuse perform equally well on non-human or abstract LoRAs (e.g., style or object-based ones)? Are the masks still meaningful in such cases?
2. The masks are computed once (step 6) and reused. How stable are these spatial associations across later denoising steps? Would re-computation improve consistency?
3. Can the authors report actual runtime and memory overhead compared to baseline inference? The claim of “one-step extraction” suggests efficiency, but quantitative figures would help.
4. Have the authors explored hierarchical or recursive masking strategies for cases with more than three subjects? Such an extension could broaden the method’s applicability.

---

> ### Author Response · Authors · 2025-11-24
> **Rebuttal from Authors of Paper8708 to Reviewer 54Vm**
>
> We sincerely appreciate Reviewer’s attention to the real-world applicability and strengths of our method. We address the concerns as follows:
>
> **Regarding W1 & Q4(The scalibility, failure analysis and possible improvements).**
>
>  In the revised version, we substantially expand scalability evaluations. The teaser, **Figure 6**, and **Figure 11** illustrate the fusion of multiple characters and objects, and **Table 3** further quantifies scalability. Our method performs reliably even when four LoRAs are applied simultaneously. In **Appendix G**, we additionally analyze failure cases: cross-attention maps indicate that, when the scene becomes overly complex, the DiT model may hallucinate and incorrectly place multiple subjects at the same position, leading to duplicated or missing characters. We discuss potential solutions, such as requiring the user to provide regional prompts or training a lightweight correction network to predict subject locations. Recursive masking may also help address such failures. We also believe that as stronger base models with better scene understanding emerge, the upper bound of our method will continue to improve.
>
> **Regarding W2(Analysis introduces some assumptions).**
>
> Our goal is not to provide a rigorous mathematical proof, but rather to offer an intuitive analysis demonstrating that applying masks effectively mitigates feature conflicts between LoRAs. We have revised our analysis. In the task of generating $N$ subjects with $N$ LoRAs applied jointly, direct joint inference introduces two forms of LoRA interference. First, in the Attention-V and FF layers, for the region $R_i$ corresponding to subject i, all LoRA outputs are summed, causing direct conflicts. Second, each LoRA’s output may be aggregated into the tokens within $R_i$ through attention, further introducing cross-LoRA interference.
>  To address this, we first apply masks to ensure that only LoRA i contributes within region $R_i$ in the FF and V pathways. Then, leveraging the well-documented locality of attention, we argue that tokens inside $R_i$ primarily attend to themselves, so the aggregation step does not bring substantial features from other LoRAs into $R_i$.
>  Consequently, our method remains simple while effectively mitigating cross-LoRA feature conflicts.
> Please refer to the updated **section “Masking LoRA Outputs for Effective Subject Feature Preservation.”**
>
> **Regarding W3(Compare with other multi LoRA based DiT method).**
>
>  To the best of our knowledge, no open-source LoRA-based multi-subject method has yet been implemented on DiT architectures.
>
> **Regarding Q1(Results on Style LoRA, Objects LoRA, Cartoon character LoRA and Real character LoRA):**
>
>  To further validate effectiveness beyond realistic human LoRAs, we conducted an expanded study including 35 character LoRAs (cartoon, 3D, anime) and 8 object LoRAs. Results show consistently strong fusion across cartoon, 3D, real-person, and object LoRAs. Please refer to the revised Teaser, **Figure 6**, **Figure 11**, and **Table 3** for details.
>
> Regarding style LoRAs, they can be seamlessly integrated by simply applying an all-ones mask. Unlike subject LoRAs, style LoRAs mainly induce stylistic transformations rather than feature conflicts, so no additional masking is required. We demonstrate diverse style-LoRA applications in the teaser, **Figure 6**, and **Figure 12**.
>
> **Regarding Q2(The reason of only computing on single timestep and single block).**
>
>  We understand the reviewer’s interest in thoroughly comparing our mask-extraction strategy to alternatives.
> We further conducted ablation studies showing that the timestep and block from which we extract the masks correspond to the model’s most accurate cross-attention maps. Recomputing them at other stages not only increases computational cost but also leads to reduced accuracy. We find that DiT models typically settle the global composition within the first 4–7 timesteps, making this window optimal for mask extraction. In terms of layer selection, only the last double-stream blocks reliably produce structured cross-attention maps; earlier blocks do not. Therefore, averaging attention across timesteps and layers, i.e., “simple averaging of attention maps (pixel queries × subject-token keys)”, significantly increases computation while reducing mask quality (see **Figure 8b**).
>
> **Regarding Q3(Actual runtime and efficiency analysis).**
>
>  We report runtime in **Appendix F**. Compared with recomputing masks at every timestep, our strategy yields ~30% speed improvement.
>  Despite running on the much heavier DiT architecture (compared with U-Net–based methods), our runtime and memory usage are comparable to or even better than OMG and CLoRA, while CLoRA incurs over 400% time overhead relative to its baseline due to its frequent attention-map computation and additional backward passes.

---

### Meta-Review · Area_Chair_XhrH · 2026-01-06

**Summary:**

This paper introducesa training-free framework for multi-subject text-to-image generation that enables fusing multiple LoRA modules without retraining. The method is implemented in two stages: (1) mask extraction via an attention maps and superpixel-based voting, and (2) mask-guided inference where LoRA outputs are masked at each denoising step.

**Reviewer Concerns:**

- Scalability to Many Subjects:
Initial submission only showed two-subject fusion.
Multiple reviewers worried performance degrades as the number of LoRAs increases (especially 5+).
Concern about shrinking masks, hallucinated overlaps, and identity leakage in crowded scenes.

- Theoretical Rigor
Requests for more direct validation of the assumptions.

- Generalization Beyond Human Characters

- Early Evaluations focused heavily on photorealistic human pairs.

- Claims of efficiency initially lacked clear comparative numbers.
Reviewers wanted concrete runtime/memory comparisons with CLoRA, OMG, etc.

**Reviewer Scores:**

Reviewer 54Vm borderline reject: the reviewer is likely to increase the score here because of the added experiments
Reviewer xeE2 boarderline accept: the reviewer is likely to increase or maintain the score
Reviewer nDjA reject: the reviewer is not fully convinced, the rebuttal does not seem to fully address the concern regarding generalisability there.
Reviewer hgmv boraderline reject: the reviwer's concern on scalability too, I won' believe that fully address the concern.

In all, the scalability concern would remain, since the added results in table 3 is still mainly charaters. Even with considering some uplifting in the scoring, this paper would still be at the borderline reject side.

---

### Decision · Program_Chairs · 2026-01-26

Reject